# Anytime Model Selection in Linear Bandits

**Parnian Kassraie**[1]     **Nicolas Emmenegger**[1]     **Andreas Krause**[1]     **Aldo Pacchiano**[2,3]

[1]ETH Zurich     [2]Broad Institute of MIT and Harvard     [3]Boston University

{pkassraie, nicolaem, krausea}@ethz.ch     pacchian@bu.edu

## Abstract

Model selection in the context of bandit optimization is a challenging problem, as it requires balancing exploration and exploitation not only for action selection, but also for model selection. One natural approach is to rely on online learning algorithms that treat different models as experts. Existing methods, however, scale poorly ($\mathrm{poly}M$) with the number of models $M$ in terms of their regret. Our key insight is that, for model selection in linear bandits, we can emulate full-information feedback to the online learner with a favorable bias-variance trade-off. This allows us to develop ALEXP, which has an exponentially improved ($\log M$) dependence on $M$ for its regret. ALEXP has anytime guarantees on its regret, and neither requires knowledge of the horizon $n$, nor relies on an initial purely exploratory stage. Our approach utilizes a novel time-uniform analysis of the Lasso, establishing a new connection between online learning and high-dimensional statistics.

## 1   Introduction

When solving bandit problems or performing Bayesian optimization, we need to commit to a reward model *a priori*, based on which we estimate the reward function and build a policy for selecting the next action. In practice, there are many ways to model the reward by considering different feature maps or hypothesis spaces, e.g., for optimizing gene knockouts [Gonzalez et al., 2015, Pacchiano et al., 2022] or parameter estimation in biological dynamical systems [Ulmasov et al., 2016, Imani et al., 2019]. It is not known a priori which model is going to yield the most sample efficient bandit algorithm, and we can only select the right model as we gather empirical evidence. This leads us to ask, can we perform adaptive model selection, while simultaneously optimizing for reward?

In an idealized setting with no sampling limits, given a model class of size $M$, we could initialize $M$ bandit algorithms (a.k.a agents) in parallel, each using one of the available reward models. Then, as the algorithms run, at every step we can select the most promising agent, according to the cumulative rewards that are obtained so far. Model selection can then be cast into an online optimization problem on a $M$-dimensional probability simplex, where the probability of selecting an agent is dependent on its cumulative reward, and the optimizer seeks to find the distribution with the best return in hindsight. This approach is impractical for large model classes, since at every step, it requires drawing $M$ different samples in parallel from the environment so that the reward for each agent is realized.

In realistic applications of bandit optimization, we can only draw *one* sample at a time, and so we need to design an algorithm which allocates more samples to the agents that are deemed more promising. Prior work [e.g., Maillard and Munos, 2011, Agarwal et al., 2017] propose to run a single meta algorithm which interacts with the environment by first selecting an agent, and then selecting an action according to the suggestion of that agent. The online model selection problem can still be emulated in this setup, however this time the optimizer receives partial feedback, coming from only one agent. Consequently, many agents need to be queried, and the overall regret scales with $\mathrm{poly}M$, again restricting this approach to small model classes. In fact, addressing the limited scope of such algorithms, Agarwal et al. [2017] raise an open problem on the feasibility of obtaining a $\log M$ dependency for the regret.

We show that this rate is achievable, in the particular case of linearly parametrizable rewards. We develop a technique to "hallucinate" the reward for every agent that was not selected, and run the online optimizer with emulated full-information feedback. This allows the optimizer to assess the quality of the agents, without ever having queried them. As a result, our algorithm ALEXP, satisfies a regret of rate $\mathcal{O}(\max\{\sqrt{n \log^3 M}, n^{3/4}\sqrt{\log M}\})$, with high probability, simultaneously for all $n \geq 1$ (Theorem 1). Our key idea, leading to $\log M$ dependency, is to employ the Lasso as a low-variance online regression oracle, and estimate the reward for the agents that were not chosen. This trick is made possible through our novel time-uniform analysis of online Lasso regression (Theorem 3). Consequently, ALEXP is horizon-independent, and explores adaptively without requiring an initial exploration stage. Empirically we find that ALEXP consistently outperforms prior work across a range of environments.

Table 1: Overview of literature on online model selection for bandit optimization

| | MS Technique | Regret | MS Guarantee | adaptive exploration | anytime |
|---|---|---|---|---|---|
| Sparse Linear Bandits | Lasso | $\log M$ | ✗ | ✗ | ✗ |
| MS for Black-Box Bandits | OMD with bandit feedback | $\text{poly}M$ | ✓ | ✓ | ✗ |
| MS for Linear Bandits (Ours) | EXP4 with full-information | $\log M$ | ✓ | ✓ | ✓ |

## 2 Related Work

Online Model selection (MS) for bandits considers combining a number of agents in a master algorithm, with the goal of performing as well as the best agent [Maillard and Munos, 2011, Agarwal et al., 2017, Pacchiano et al., 2020, Luo et al., 2022]. This literature operates on black-box model classes of size $M$ and uses variants of Online Mirror Descent (OMD) to sequentially select the agents. The optimizer operates on importance-weighted estimates of the agents' rewards, which has high variance ($\text{poly}M$) and is non-zero only for the selected agent. Effectively, the optimizer receives partial (a.k.a. bandit) feedback and agents are at risk of starvation, since at every step only the selected agent gets the new data point. These works assume knowledge of the horizon, however, we suspect that this may be lifted with a finer analysis of OMD.

Sparse linear bandits use sparsity-inducing methods, often Lasso [Tibshirani, 1996], for estimating the reward in presence of many features, as an alternative to model selection. Early results are in the data-rich regime where the stopping time $n$ is known and larger than the number of models, and some suffer from $\text{poly}M$ regret dependency [e.g., Abbasi-Yadkori et al., 2012]. Recent efforts often consider the contextual case, where at every step only a finite stochastic subset of the action domain is presented to the agent, and that the distribution of these points is i.i.d. and sufficiently diverse [Li et al., 2022, Bastani and Bayati, 2020, Kim and Paik, 2019, Oh et al., 2021, Cella and Pontil, 2021]. We do not rely on such assumptions. Most sparse bandit algorithms either start with a purely exploratory phase [Kim and Paik, 2019, Li et al., 2022, Hao et al., 2020, Jang et al., 2022], or rely on a priori scheduled exploration [Bastani and Bayati, 2020]. The exploration budget is set according to the horizon $n$. Therefore, such algorithms inherently require the knowledge of $n$ and can be made anytime only via the doubling trick [Auer et al., 1995]. Table 2 presents an in-depth overview.

ALEXP inherits the best of both worlds (Table 1): its regret enjoys the $\log M$ dependency of sparse linear bandits even on compact domains, and it has adaptive probabilistic exploration with anytime guarantees. In contrast to prior literature, we perform model selection with an online optimizer (EXP4), which hallucinates full-information feedback using a low-variance Lasso estimator instead of importance-weighted estimates. Moreover, our anytime approach lifts the horizon dependence and the exploration requirement of sparse linear bandits.

Our work is inspired by and contributes to the field of Learning with Expert Advice, which analyzes incorporating the advice of $M$ oblivious (non-adaptive) experts, with bandit or full-information feedback [Haussler et al., 1998, Auer et al., 2002b, McMahan and Streeter, 2009]. The idea of employing an online optimizer for learning stems from this literature, and has been used in various applications of online learning [Foster et al., 2017, Singla et al., 2018, Muthukumar et al., 2019, Karimi et al., 2021, Liu et al., 2022]. In particular, we are inspired by Foster and Rakhlin [2020] and Moradipari et al. [2022], who apply EXP4 to least squares estimates, for arm selection in $K$-armed

contextual bandits. However, their algorithms are not anytime, and due to the choice of estimator, the corresponding regret scales with $\mathcal{O}(\min(\sqrt{M}, \sqrt{K \log M}))$.

# 3 Problem Setting

We consider a bandit problem where a learner interacts with the environment in rounds. At step $t$ the learner selects an action $\boldsymbol{x}_t \in \mathcal{X}$, where $\mathcal{X} \subset \mathbb{R}^{d_0}$ is a compact domain and observes a noisy reward $y_t = r(\boldsymbol{x}_t) + \varepsilon_t$ such that $\varepsilon_t$ is an i.i.d. zero-mean sub-Gaussian variable with parameter $\sigma^2$. We assume the reward function $r : \mathcal{X} \to \mathbb{R}$ is linearly parametrizable by some unknown feature map, and that the model class $\{\boldsymbol{\phi}_j : \mathbb{R}^{d_0} \to \mathbb{R}^d, j = 1, \ldots, M\}$ contains the set of plausible feature maps. We consider the setting where $M$ can be very large, and while the set $\{\boldsymbol{\phi}_j\}$ may include misspecified feature maps, it contains at least one feature map that represents the reward function, i.e., there exists $j^\star \in [M]$ such that $r(\cdot) = \boldsymbol{\theta}_{j^\star}^\top \boldsymbol{\phi}_{j^\star}(\cdot)$. We assume the image of $\boldsymbol{\phi}_j$ spans $\mathbb{R}^d$, and no two feature maps are linearly dependent, i.e. for any $j, j' \in [M]$, there exists no $\alpha \in \mathbb{R}$ such that $\boldsymbol{\phi}_j(\cdot) = \alpha \boldsymbol{\phi}_{j'}(\cdot)$. This assumption, which is satisfied by design in practice, ensures that the features are not ill-posed and we can explore in all relevant directions. We assume that the concatenated feature map $\boldsymbol{\phi}(\boldsymbol{x}) := (\boldsymbol{\phi}_1(\boldsymbol{x}), \ldots, \boldsymbol{\phi}_M(\boldsymbol{x}))$ is normalized $\|\boldsymbol{\phi}(\boldsymbol{x})\| \leq 1$ for all $\boldsymbol{x} \in \mathcal{X}$ and that $\|\boldsymbol{\theta}_{j^\star}\| \leq B$, which implies $|r(\boldsymbol{x})| \leq B$ for all $\boldsymbol{x} \in \mathcal{X}$.

We will model this problem in the language of model selection where a meta algorithm aims to optimize the unknown reward function by relying on a number of base learners. In order to interact with the environment the meta algorithm selects an agent that in turn selects an action. In our setting we thinking of each of these $M$ feature maps as controlled by a base agent running its own algorithm. Base agent $j$ uses the feature map $\boldsymbol{\phi}_j$ for modeling the reward. At step $t$ of the bandit problem, each agent $j$ is given access to the full history $H_{t-1} := \{(\boldsymbol{x}_1, y_1), \ldots, (\boldsymbol{x}_{t-1}, y_{t-1})\}$, and uses it to locally estimate the reward as $\hat{\boldsymbol{\beta}}_{t-1,j}^\top \boldsymbol{\phi}_j(\cdot)$, where $\hat{\boldsymbol{\beta}}_{t-1,j} \in \mathbb{R}^d$ is the estimated coefficients vector. The agent then uses this estimate to develop its action selection policy $p_{t,j} \in \mathcal{M}(\mathcal{X})$. Here, $\mathcal{M}$ denotes the space of probability measures defined on $\mathcal{X}$. The condition on existence of $j^\star$ will ensure that there is at least one agent which is using a correct model for the reward, and thereby can solve the bandit problem if executed in isolation. We refer to agent $j^\star$ as the oracle agent.

Our goal is to find a sample-efficient strategy for iterating over the agents, such that their suggested actions maximize the cumulative reward, achieved over any horizon $n \geq 1$. This is equivalent to minimizing the cumulative regret $R(n) = \sum_{t=1}^{n} r(\boldsymbol{x}^*) - r(\boldsymbol{x}_t)$, where $\boldsymbol{x}^*$ is a global maximizer of the reward function. We neither fix $n$, nor assume knowledge of it.

# 4 Method

As warm-up, consider an example with deterministic agents, i.e., when $p_{t,j}$ is a Dirac measure on a specific action $\boldsymbol{x}_{t,j}$. Suppose it was practically feasible to draw the action suggested by every agent and observe the corresponding reward vector $\boldsymbol{r}_t = (r_{t,j})_{j=1}^{M}$. In this case, model selection becomes a full-information online optimization problem, and we can design a minimax optimal algorithm as follows. We assign a probability distribution $\boldsymbol{q}_t = (q_{t,j})_{j=1}^{M}$ to the models, and update it such that the overall average return $\sum_{t=1}^{n} \boldsymbol{q}_t^\top \boldsymbol{r}_t$ is competitive to the best agent's average return $\sum_{t=1}^{n} r_{t,j^\star}$. At every step, we update $q_{t+1,j} \propto \exp(\sum_{s=1}^{t} r_{s,j})$, since such exponential weighting is known to lead to an optimal solution for this classic online learning problem [Cesa-Bianchi and Lugosi, 2006]. In our setting however, the agents are stochastic, and we do not have access to the full $\boldsymbol{r}_t$ vector.

We propose the **A**nytime **EXP**onential weighting algorithm based on **L**asso reward estimates (ALEXP), summarized in Algorithm 1. At step $t$ we first sample an agent $j_t$, and then sample an action $\boldsymbol{x}_t$ according to the agent's policy $p_{t,j_t}$. Let $\Delta_M$ denote the $M$-dimensional probability simplex. We maintain a probability distribution $\boldsymbol{q}_t \in \Delta_M$ over the agents, and update it sequentially as we accumulate evidence on the performance of each agent. Ideally, we would have adjusted $q_{t,j}$ according to the average return of model $j$, that is, $\mathbb{E}_{\boldsymbol{x} \sim p_{t,j}} r(\boldsymbol{x})$. However, since $r$ is unknown, we estimate the average reward with some $\hat{r}_{t,j}$. We then update $\boldsymbol{q}_t$ for the next step via,

$$q_{t+1,j} = \frac{\exp(\eta_t \sum_{s=1}^{t} \hat{r}_{s,j})}{\sum_{i=1}^{M} \exp(\eta_t \sum_{s=1}^{t} \hat{r}_{s,i})}$$

for all $j = 1, \ldots, M$, where $\eta_t$ is the learning rate, and controls the sensitivity of the updates. This rule allows us to imitate the full-information example that we mentioned above. By utilizing $\hat{r}_{t,j}$ and hallucinating feedback from all agents, we can reduce the probability of selecting a badly performing agent, without ever having sampled them (c.f. Fig. 4). It remains to design the estimator $\hat{r}_{t,j}$. We concatenate all feature maps, and, knowing that many features are redundant, use a sparsity inducing estimator over the resulting coefficients vector. Mainly, let $\boldsymbol{\theta} = (\boldsymbol{\theta}_1, \ldots, \boldsymbol{\theta}_M) \in \mathbb{R}^{Md}$ be the concatenated coefficients vector. We then solve

$$\hat{\boldsymbol{\theta}}_t = \underset{\boldsymbol{\theta} \in \mathbb{R}^{Md}}{\arg\min} \mathcal{L}(\boldsymbol{\theta}; H_t, \lambda_t) = \underset{\boldsymbol{\theta} \in \mathbb{R}^{Md}}{\arg\min} \frac{1}{t} \|\boldsymbol{y}_t - \Phi_t \boldsymbol{\theta}\|_2^2 + 2\lambda_t \sum_{j=1}^{M} \|\boldsymbol{\theta}_j\|_2 \tag{1}$$

where $\Phi_t = [\boldsymbol{\phi}^\top(\boldsymbol{x}_s)]_{s \leq t} \in \mathbb{R}^{t \times Md}$ is the feature matrix, $\boldsymbol{y}_t \in \mathbb{R}^t$ is the concatenated reward vector, and $\lambda_t$ is an adaptive regularization parameter. Problem (1) is the online variant of the group Lasso [Lounici et al., 2011]. The second term is the loss is the mixed $(2, 1)$-norm of $\boldsymbol{\theta}$, which can be seen as the $\ell_1$-norm of the vector $(\|\boldsymbol{\theta}_1\|, \ldots, \|\boldsymbol{\theta}_M\|) \in \mathbb{R}^M$. This norm induces sparsity at the group level, and therefore, the sub-vectors $\boldsymbol{\theta}_{t,j} \in \mathbb{R}^d$ that correspond to redundant feature maps are expected to be $\boldsymbol{0}$, i.e. the null vector. We then estimate the average return of each model by simply taking an expectation $\hat{r}_{t,j} = \mathbb{E}_{\boldsymbol{x} \sim p_{t+1,j}}[\hat{\boldsymbol{\theta}}_t^\top \boldsymbol{\phi}(\boldsymbol{x})]$. This quantity is the average return of the agent's policy $p_{t+1,j}$, according to our Lasso estimator. In Section 5.2 we explain why the particular choice of Lasso is crucial for obtaining a $\log M$ rate for the regret.

For action selection, with probability $\gamma_t$, we sample agent $j$ with probability $q_{t,j}$ and draw $\boldsymbol{x}_t \sim p_{t,j}$ as per suggestion of the agent. With probability $1 - \gamma_t$, we choose the action according to some exploratory distribution $\pi \in \mathcal{M}(\mathcal{X})$ which aims to sample informative actions. This can be any design where $\text{supp}(\pi) = \mathcal{X}$. We mix $p_{t,j}$ with $\pi$, to collect sufficiently diverse data for model selection. We are not restricting the agents' policy, and therefore can not rely on them to explore adequately. In Theorem 1, we choose a decreasing sequence of $(\gamma_t)_{t \geq 1}$ the probabilities of exploration at step $t \geq 1$, since less exploration will be required as data accumulates. To conclude, the action selection policy of ALEXP is formally described as the mixture

$$p_t(\boldsymbol{x}) = \gamma_t \pi(\boldsymbol{x}) + (1 - \gamma_t) \sum_{j=1}^{M} q_{t,j} p_{t,j}(\boldsymbol{x}).$$

## 5 Main Results

For the regret guarantee, we consider specific choices of base agents and exploratory distribution. Our analysis may be extended to include other policies, since ALEXP can be wrapped around any bandit agent that is described by some $p_{t,j}$, and allows for random exploration with any distribution $\pi$.

**Base Agents.** We assume that the oracle agent has either a UCB [Abbasi-Yadkori et al., 2011] or a GREEDY [Auer et al., 2002a] policy, and all other agents are free to choose *any arbitrary* policy. Similar treatment can be applied to cases where the oracle uses other (sublinear) polices for solving linear or contextual bandits [e.g., Thompson, 1933, Kaufmann et al., 2012, Agarwal et al., 2014]. In either case, agent $j^\star$ calculates a ridge estimate of the coefficients vector based on the history $H_t$

$$\hat{\boldsymbol{\beta}}_{t,j^\star} \coloneqq \underset{\boldsymbol{\beta} \in \mathbb{R}^d}{\arg\min} \|\boldsymbol{y}_t - \Phi_{t,j^\star} \boldsymbol{\beta}\|_2^2 + \tilde{\lambda} \|\boldsymbol{\beta}\|_2^2 = \left( \Phi_{t,j^\star}^\top \Phi_{t,j^\star} + \tilde{\lambda} \boldsymbol{I} \right)^{-1} \Phi_{t,j^\star}^\top \boldsymbol{y}_t.$$

Here $\Phi_{t,j^\star} \in \mathbb{R}^{t \times d}$ is the feature matrix, where each row $s$ is $\boldsymbol{\phi}_{j^\star}^\top(\boldsymbol{x}_s)$ and $\tilde{\lambda}$ is the regularization constant. Then at step $t$, a GREEDY oracle suggests the action which maximizes the reward estimate $\hat{\boldsymbol{\beta}}_{t-1,j^\star}^\top \boldsymbol{\phi}_{j^\star}(\boldsymbol{x})$, and a UCB oracle queries $\arg\max u_{t-1,j^\star}(\boldsymbol{x})$ where $u_{t-1,j^\star}(\cdot)$ is the upper confidence bound that this agent calculates for $r(\cdot)$. Proposition 21 shows that the sequence $(u_{t,j^\star})_{t \geq 1}$ is in fact an anytime valid upper bound for $r$ over the entire domain.

**Exploratory policy.** Performance of ALEXP depends on the quality of the samples that $\pi$ suggests. The eigenvalues of the covariance matrix $\Sigma(\pi, \boldsymbol{\phi}) \coloneqq \mathbb{E}_{\boldsymbol{x} \sim \pi} \boldsymbol{\phi}(\boldsymbol{x}) \boldsymbol{\phi}^\top(\boldsymbol{x})$ reflect how diverse the data is, and thus are a good indicator for data quality. van de Geer and Bühlmann [2009] present a survey on the notions of diversity defined based on these eigenvalues. Let $\lambda_{\min}(A)$ denote the minimium eigenvalue of a matrix $A$. Similar to Hao et al. [2020], we assume that $\pi$ is the maximizer of the problem below and present our regret bounds in terms of

$$C_{\min} = C_{\min}(\mathcal{X}, \boldsymbol{\phi}) \coloneqq \max_{\pi \in \mathcal{M}(\mathcal{X})} \lambda_{\min}(\Sigma(\pi, \boldsymbol{\phi})), \tag{2}$$

---

**Algorithm 1** ALEXP

---

Inputs: $\pi, (\gamma_t, \eta_t, \lambda_t)$ for $t \geq 1$
Let $\boldsymbol{q}_1 \leftarrow \text{Unif}(M)$ and initialize base agents $(p_{1,1}, \ldots, p_{1,M})$.
**for** $t \geq 1$ **do**
    Draw $\alpha_t \sim \text{Bernoulli}(\gamma_t)$.        ▷ Decide to explore or exploit
    **if** $\alpha_t = 1$ **then**
        Choose action $\boldsymbol{x}_t$ randomly according to $\pi$.        ▷ Explore
    **else**
        Draw $j_t \sim \boldsymbol{q}_t$.        ▷ Select an agent
        Draw $\boldsymbol{x}_t \sim p_{t,j_t}$.        ▷ Select the action suggested by the agent
    **end if**
    Observe $y_t = r(\boldsymbol{x}_t) + \varepsilon_t$.        ▷ Receive reward
    $H_t = H_{t-1} \cup \{(\boldsymbol{x}_t, y_t)\}$.        ▷ Append history
    $\hat{\boldsymbol{\theta}}_t \leftarrow \arg\min \mathcal{L}(\boldsymbol{\theta}; H_t, \lambda_t)$.        ▷ Update the parameter estimate
    Report $H_t$ to all agents, and get updated policies $p_{t+1,1}, \ldots, p_{t+1,M}$.        ▷ Update agents
    Update estimated average return of every agent

$$\hat{r}_{t,j} \leftarrow \mathbb{E}_{\boldsymbol{x} \sim p_{t+1,j}}[\hat{\boldsymbol{\theta}}_t^\top \boldsymbol{\phi}(\boldsymbol{x})], \quad j = 1, \ldots, M$$

    Update agent probabilities

$$q_{t+1,j} \leftarrow \frac{\exp(\eta_t \sum_{s=1}^{t} \hat{r}_{s,j})}{\sum_{i=1}^{M} \exp(\eta_t \sum_{s=1}^{t} \hat{r}_{s,i})}$$

**end for**

---

which is greater than zero under the conditions specified in our problem setting. Prior works in the sparse bandit literature all require a similar or stronger assumption of this kind, and Table 2 gives an overview. Alternatively, one can work with an arbitrary $\pi$, e.g., $\text{Unif}(\mathcal{X})$, as long as $\lambda_{\min}(\Sigma(\pi, \boldsymbol{\phi}))$ is bounded away from zero. Appendix C.1 reviews some configurations of $(\boldsymbol{\phi}, \mathcal{X}, \pi)$ that lead to a non-zero minimum eigenvalue, and Corollary 12 bounds the regret of ALEXP with uniform exploration.

For this choice of agents and exploratory distribution, Theorem 1 presents an informal regret bound. Here, we have used the $\mathcal{O}$ notation, and only included the fastest growing terms. The inequality is made exact in Theorem 14, up to constant multiplicative factors.

**Theorem 1** (Cumulative Regret of ALEXP, Informal). *Let $\delta \in (0, 1]$ and set $\pi$ to be the maximizer of* (2). *Choose learning rate $\eta_t = \mathcal{O}(C_{\min} t^{-1/2} / C(M, \delta, d))$, exploration probability $\gamma_t = \mathcal{O}(t^{-1/4})$ and Lasso regularization parameter $\lambda_t = \mathcal{O}(C_{\min} t^{-1/2} C(M, \delta, d))$, where*

$$C(M, \delta, d) = \mathcal{O}\left(\sqrt{1 + \log(M/\delta) + (\log\log d)_+} + \sqrt{d\left(\log(M/\delta) + (\log\log d)_+\right)}\right).$$

*Then* ALEXP *satisfies the cumulative regret*

$$R(n) = \mathcal{O}\left(n^{3/4}B + n^{3/4}C(M, \delta, d) + C_{\min}^{-1}\sqrt{n}C(M, \delta, d)\log M \right.$$
$$\left. + C_{\min}^{-1/2}n^{5/8}\sqrt{d\log(n) + \log(1/\delta)}\right)$$

*simultaneously for all $n \geq 1$, with probability greater than $1 - \delta$.*

In this bound, the first term is the regret incurred at exploratory steps (when $\alpha_t = 1$), the second term is due to the estimation error of Lasso (i.e., $\|\boldsymbol{\theta} - \hat{\boldsymbol{\theta}}_t\|$), and the third term is the regret of the exponential weights sub-algorithm. The fourth term, is the regret bound for the oracle agent $j^\star$, when run within the ALEXP framework. It does not depend on the agent's policy (greedy or optimistic), and is worse than the minimax optimal rate of $\sqrt{nd\log n}$. This is because the oracle is suggesting actions based on the history $H_t$, which includes uninformative action-reward pairs queried by other, potentially misspecified, agents. In Corollary 12, we provide a regret bound independent of $C_{\min}$, for orthogonal feature maps, and show that the same $\mathcal{O}(\max\{\sqrt{n\log^3 M}, n^{3/4}\sqrt{\log M}\})$ rate may be achieved even with the simple choice $\pi = \text{Unif}(\mathcal{X})$.

## 5.1 Proof Sketch

The regret is caused by two sources: selecting a sub-optimal agent, and an agent selecting a sub-optimal action. Accordingly, for any $j \in 1, \ldots, M$, we decompose the regret as

$$R(n) = \sum_{t=1}^{n} r(\boldsymbol{x}^{\star}) - r(\boldsymbol{x}_t) = \left( \sum_{t=1}^{n} r(\boldsymbol{x}^{\star}) - r_{t,j} \right) + \left( \sum_{t=1}^{n} r_{t,j} - r(\boldsymbol{x}_t) \right). \tag{3}$$

The first term shows the cumulative regret of agent $j$, when run within ALEXP. The second term evaluates the received reward against the cumulative average reward of model $j$. We bound each term separately.

**Virtual Regret.** The first term $\tilde{R}_j(n) := \sum_{t=1}^{n} r(\boldsymbol{x}^{\star}) - r_{t,j}$ compares the suggestion of agent $j$, against the optimal action. We call it the virtual regret since the sequence $(\boldsymbol{x}_{t,j})_{t \geq 1}$ of the actions suggested by model $j$ are not necessarily selected by the meta algorithm, unless $j_t = j$. This regret is merely a technical tool, and not actually realized when running ALEXP. The virtual regret of the oracle agent may still be analyzed using standard techniques for linear bandits, e.g., Abbasi-Yadkori et al. [2011], however we need to adapt it to take into account a subtle difference: The confidence sequence of model $j^{\star}$ is constructed according to the true sequence of actions $(\boldsymbol{x}_t)_{t \geq 1}$, while its virtual regret is calculated based on the virtual sequence $(\boldsymbol{x}_{t,j^{\star}})_{t \geq 1}$, which the model suggests. The two sequences only match at the steps when model $j^{\star}$ is selected. Adapting the analysis of Abbasi-Yadkori et al. [2011] to this subtlety, we obtain in Lemma 15 that with probability greater than $1 - \delta$, simultaneously for all $n \geq 1$,

$$\tilde{R}_{j^{\star}}(n) = \mathcal{O}\left( n^{5/8} C_{\min}^{-1/2} \sqrt{d \log(n) + \log(1/\delta)} \right).$$

**Model Selection Regret.** The second term in (3) is the model selection regret, $R(n, j) := \sum_{t=1}^{n} r_{t,j} - r(\boldsymbol{x}_t)$, which evaluates the chosen action by ALEXP against the suggestion of the $j$-th agent. Our analysis relies on a careful decomposition of $R(n, j)$,

$$R(n, j) = \sum_{t=1}^{n} \left[ \underbrace{r_{t,j} - \hat{r}_{t,j}}_{(I)} + \underbrace{\hat{r}_{t,j} - \sum_{i=1}^{M} q_{t,i} \hat{r}_{t,i}}_{(II)} + \underbrace{\sum_{i=1}^{M} q_{t,i}(\hat{r}_{t,i} - r_{t,i})}_{(III)} + \underbrace{\sum_{i=1}^{M} q_{t,i} r_{t,i} - r(\boldsymbol{x}_t)}_{(IV)} \right].$$

We bound away each term in a modular manner, until we are left with the regret of the standard exponential weights algorithm. The terms (I) and (III) are controlled by the bias of the Lasso estimator, and are $\mathcal{O}(n^{3/4} C(M, \delta, d))$ (Lemma 19). The last term (IV) is zero in expectation, and reflects the deviation of $r(\boldsymbol{x}_t)$ from its mean. We observe that the summands form a bounded Martingale difference sequence, and their sum grows as $\mathcal{O}(\sqrt{n})$ (Lemma 18). Term (II) is the regret of our online optimizer, which depends on the variance of the Lasso estimator. We bound this term with $\mathcal{O}(\sqrt{n} C(M, \delta, d) \log M)$, by first conducting a standard anytime analysis of exponential weights (Lemma 17), and then incorporating the anytime Lasso variance bound (Lemma 20). We highlight that neither of the above steps require assumptions about the base agents. Combining these steps, Lemma 16 establishes the formal bound on the model selection regret.

**Anytime Lasso.** We develop novel confidence intervals for Lasso with history-dependent data, which are uniformly valid over an unbounded time horizon. This result may be of independent interest in applications of Lasso for online learning or sequential decision making. Here, we use these confidence intervals to bound the bias and variance terms that appear in our treatment of the model selection regret. The width of the Lasso confidence intervals depends on the quality of feature matrix $\Phi_t$, often quantified by the restricted eigenvalue property [Bickel et al., 2009, van de Geer and Bühlmann, 2009, Javanmard and Montanari, 2014]:

**Definition 2.** For the feature matrix $\Phi_t \in \mathbb{R}^{t \times dM}$ we define $\kappa(\Phi_t, s)$ for any $1 \leq s \leq M$ as

$$\kappa(\Phi_t, s) := \inf_{(J, \boldsymbol{b})} \frac{1}{\sqrt{t}} \frac{\|\Phi_t \boldsymbol{b}\|_2}{\sqrt{\sum_{j \in J} \|\boldsymbol{b}_j\|_2^2}}$$

$$\text{s.t. } \boldsymbol{b} \in \mathbb{R}^d \setminus \{0\}, \sum_{j \notin J} \|\boldsymbol{b}_j\|_2 \leq 3 \sum_{j \in J} \|\boldsymbol{b}_j\|_2, J \subset \{1, \ldots, M\}, |J| \leq s.$$

Our analysis is in terms of this quantity, and Lemma 8 explains the connection between $\kappa(\Phi_t, s)$ and $C_{\min}$ as defined in (2), particularly that $\kappa(\Phi_t, 2)$ is also positive with a high probability.

**Theorem 3** (Anytime Lasso Confidence Sequences). *Consider the data model $y_t = \boldsymbol{\theta}^\top \boldsymbol{\phi}(\boldsymbol{x}_t) + \varepsilon_t$ for all $t \geq 1$, where $\varepsilon_t$ is i.i.d. zero-mean sub-Gaussian noise, and $(\boldsymbol{x}_t)_{t \geq 1}$ is $(\mathcal{F}_t)_{t \geq 1}$-predictable, where $\mathcal{F}_t := (\boldsymbol{x}_1, \ldots, \boldsymbol{x}_t, \varepsilon_1, \ldots, \varepsilon_{t-1})$. Then the solution of (1) guarantees*

$$\mathbb{P}\left(\forall t \geq 1 : \left\|\boldsymbol{\theta} - \hat{\boldsymbol{\theta}}_t\right\|_2 \leq \frac{4\sqrt{10}\lambda_t}{\kappa^2(\Phi_t, 2)}\right) \geq 1 - \delta$$

*if the regularization parameter satisfies for all $t \geq 1$*

$$\lambda_t \geq \frac{2\sigma}{\sqrt{t}}\sqrt{1 + \frac{12}{\sqrt{2}}\left(\log(2M/\delta) + (\log\log d)_+\right)} + \frac{5}{\sqrt{2}}\sqrt{d\left(\log(2M/\delta) + (\log\log d)_+\right)}.$$

Our confidence bound enjoys the same rate as Lasso with fixed (offline) data, up to $\log\log d$ factors. We prove this theorem by constructing a self-normalized martingale sequence based on the $\ell_2$-norm of the empirical process error $(\Phi_t^\top \boldsymbol{\varepsilon}_t)$. We then apply the "stitched" time-uniform boundary of Howard et al. [2021]. Appendix B elaborates on this technique. Previous work on sparse linear bandits also include analysis of Lasso in an online setting, when $\boldsymbol{x}_t$ is $\mathcal{F}_t$ measurable. Cella and Pontil [2021] imitate offline analysis and then apply a union bound over the time steps, which multiplies the width by $\log n$ and requires knowledge of the horizon. Bastani and Bayati [2020] also rely on knowledge of $n$ and employ a scalar-valued Bernstein inequality on $\ell_\infty$-norm of the empirical process error, which inflates the width of the confidence sets by a factor of $\sqrt{d}$. We work directly on the $\ell_2$-norm, and use a curved boundary for sub-Gamma martingale sequences, which according to Howard et al. [2021] is uniformly tighter than a Bernstein bound, especially for small $t$.

### 5.2 Discussion

In light of Theorem 1 and Theorem 3, we discuss some properties of ALEXP.

**Sparse EXP4.** Our approach presents a new connection between online learning and high-dimensional statistics. The rule for updating the probabilities in ALEXP is inspired by the exponential weighting for Exploration and Exploitation with Experts (EXP4) algorithm, which was proposed by Auer et al. [2002b] and has been extensively studied in the adversarial bandit and learning with expert advice literature [e.g., McMahan and Streeter, 2009, Beygelzimer et al., 2011]. EXP4 classically uses importance-weighted (IW) or ordinary least squares (LS) estimators to estimate the return $r_{t,j}$, both of which are unbiased but high-variance choices [Bubeck et al., 2012]. In particular, in our linearly parametrized setting, the variance of IW and LS scales with $Md$, which will lead to a $\text{poly}(M)$ regret. However, it is known that introducing bias can be useful if it reduces the variance [Zimmert and Lattimore, 2022]. For instance, EXP3-IX [Kocák et al., 2014] and EXP4-IX [Neu, 2015] construct a biased IW estimator. Equivalently, others craft regularizers for the reward of the online optimizer, seeking to improve the bias-variance balance [e.g., Abernethy et al., 2008, Bartlett et al., 2008, Abernethy and Rakhlin, 2009, Bubeck et al., 2017, Lee et al., 2020, Zimmert and Lattimore, 2022]. A key technical observation in this work is that our online Lasso estimator leads EXP4 to achieve sublinear regret which depends logarithmically on $M$. This is due to the fact that while the estimator itself is $Md$-dimensional, its bias squared and variance scale with $\sqrt{d\log M}$. To the best of our knowledge, this work is first to instantiate the EXP4 algorithm with a sparse low-variance estimator.

**Adaptive and Anytime.** To estimate the reward, prior work on sparse bandits commonly emulate the Lasso analysis on offline data or on a martingale sequence with a known length [Hao et al., 2020, Bastani and Bayati, 2020]. These works require a long enough sequence of exploratory samples, and knowledge of the horizon to plan this sequence. ALEXP removes both of these constraints, and presents a fully adaptive algorithm. Crucially, we employ the elegant martingale bounds of Howard et al. [2020] to present the first time-uniform analysis of the Lasso with history-dependent data (Theorem 3). This way we can use all the data points and explore with a probability which vanishes at a $\mathcal{O}(t^{-1/4})$ rate. Our anytime confidence bound for Lasso, together with the horizon-independent analysis of the exponential weights algorithm, also allows ALEXP to be stopped at any time with valid guarantees.

**Rate Optimality.** For $M \gg n$, we obtain a $\mathcal{O}(\sqrt{n\log^3 M})$ regret, which matches the rate conjectured by Agarwal et al. [2017]. However, if $M$ is comparable to $n$ or smaller, our regret scales with

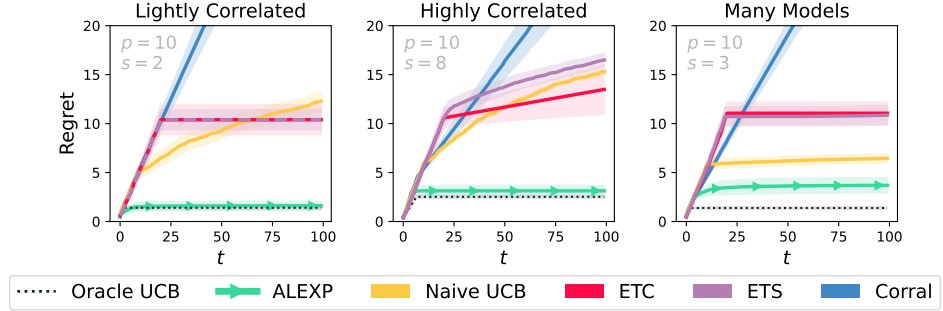

Figure 1: ALEXP can model-select in both orthogonal and correlated classes ($M = 55$)

Figure 2: ALEXP performs well on a large class ($M = 165$)

$\mathcal{O}(n^{3/4}\sqrt{\log M})$, and while it is still sublinear and scales logarithmically with $M$, the dependency on $n$ is sub-optimal. This may be due to the conservative nature of our model selection analysis, during which we do not make assumptions about the dynamics of the base agents. Therefore, to ensure sufficiently diverse data for successful model selection, we need to occasionally choose exploratory actions with a vanishing probability of $\gamma_t$. We conjecture that this is avoidable, if we make more assumptions about the agents, e.g., that a sufficient number of agents can achieve sublinear regret if executed in isolation. Banerjee et al. [2023] show that the data collected by sublinear algorithms organically satisfies a minimum eigenvalue lowerbound, which may also be sufficient for model selection. We leave this as an open question for future work.

## 6 Experiments

**Experiment Setup.** We create a synthetic dataset based on our data model (Section 3), and choose the domain to be 1-dimensional $\mathcal{X} = [-1, 1]$. As a natural choice of features, we consider the set of degree-$p$ Legendre polynomials, since they form an orthonormal basis for $L^2(\mathcal{X})$ if $p$ grows unboundedly. We construct each feature map, by choosing $s$ different polynomials from this set, and therefore obtaining $M = \binom{p+1}{s}$ different models. More formally, we let $\phi_j(x) = (P_{j_1}(x), \ldots, P_{j_s}(x)) \in \mathbb{R}^s$ where $\{j_1, \ldots, j_s\} \subset \{0, \ldots, p\}$ and $P_{j'}$ denotes a degree $j'$ Legendre polynomial. To construct the reward function, we randomly sample $j^\star$ from $[M]$, and draw $\boldsymbol{\theta}_{j^\star}$ from an i.i.d. standard gaussian distribution. We then normalize $||\boldsymbol{\theta}_{j^\star}|| = 1$. When sampling from the reward, we add Gaussian noise with standard deviation $\sigma = 0.01$. Figure 5 in the appendix shows how the random reward functions may look. For all experiments we set $n = 100$, and plot the cumulative regret $R(n)$ averaged over 20 different random seeds, the shaded areas in all figures show the standard error across these runs.

**Algorithms.** We perform experiments on two UCB algorithms, one with oracle knowledge of $j^\star$, and a naive one which takes into account all $M$ feature maps. We run Explore-then-Commit (ETC) by Hao et al. [2020], which explores for a horizon of $n_0$ steps, performs Lasso once, and then selects actions greedily for the remaining steps. As another baseline, we introduce Explore-then-Select (ETS) that explores for $n_0$ steps, performs model selection using the sparsity pattern of the Lasso estimator. For the remaining steps, the policy switches to UCB, calculated based on the selected features. Performance of ETC and ETS depends highly on $n_0$, so we tune this hyperparameter separately for each experiment. We also run CORRAL as proposed by Agarwal et al. [2017], with UCB agents similar to ALEXP. We tune the hyper-parameters of CORRAL as well. To initialize ALEXP we set the rates of $\lambda_t, \gamma_t$ and $\eta_t$ according to Theorem 1, and perform a light hyper-parameter tuning to choose the scaling constants. We have included the details and results of our hyper-parameter tuning in Appendix F.1. To solve (1), we use CELER, a fast solver for the group Lasso [Massias et al., 2018]. Every time UCB policy is used, we set the exploration coefficient $\beta_t = 2$, and every time exploration is required, we sample according to $\pi = \text{Unif}(\mathcal{X})$. Appendix F includes the pseudo-code for all baseline algorithms.[1]

**Easy vs. Hard Cases.** We construct an easy problem instance, where $s = 2$, $p = 10$, and thus $M = 55$. Models are lightly correlated since each two model can have at most one Legendre polynomial in common. We also generate an instance with highly correlated feature maps where $s = 8$ and $p = 10$, which will be a harder problem, since out of the total $M = 55$ models, there are 36 models which have at least 6 Legendre polynomials in common with the oracle model $j^\star$. Figure 1 shows that not only ALEXP is not affected by the correlations between the models, but also it achieves

---

[1]The PYTHON code for reproducing the experiments is accessible on github.com/lasgroup/ALEXP.

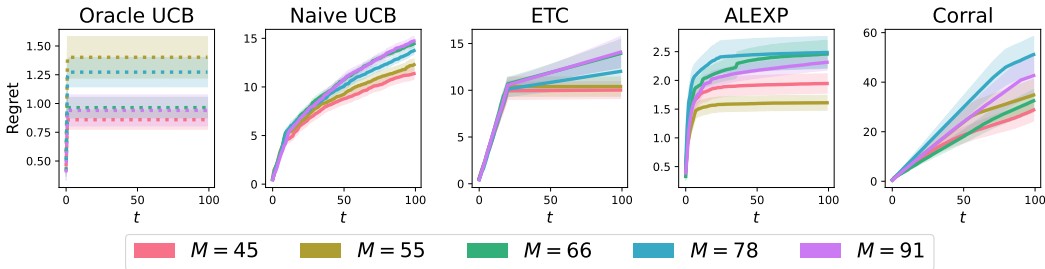

Figure 3: ALEXP is hardly affected by increasing the number of models (y-axis have various scales)

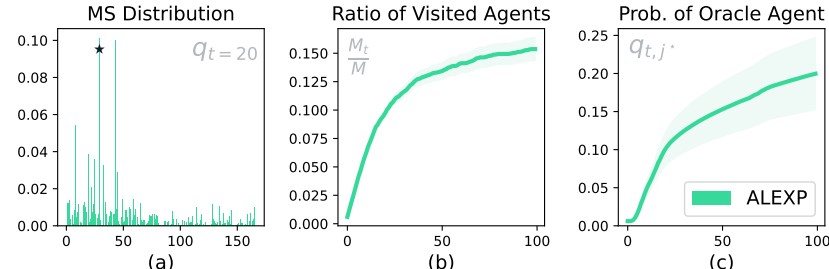

Figure 4: ALEXP can rule out models without ever having queried them ($M = 165$)

a performance competitive to the oracle in both cases, implying that our exponential weights technique for model selection is robust to choice of features. ETC and ETS rely on Lasso for model selection, which performs poorly in the case of correlated features. CORRAL uses log-barrier-OMD with an importance-weighted estimator, which has a significantly high variance. The curve for CORRAL in Figures 1 and 2 is cropped since the regret values get very large. Figure 6 shows the complete results. We construct another hard instance (Fig. 2), where the model class is large ($s = 3, p = 10, M = 165$). ALEXP continues to outperform all baselines with a significant gap. It is clear in the regret curves how explore-then-commit style algorithms are inherently horizon-dependent, and may exhibit linear regret, if stopped at an arbitrary time. This is not an issue with the other algorithms.

**Scaling with M.** Figure 3 shows how well the algorithms scale as $M$ grows. For this experiment we set $s = 2$ and change $p \in \{9, \ldots, 13\}$. While increasing $M$ hardly affects ALEXP and Oracle UCB, other baselines become less and less sample efficient.

**Learning Dynamics of ALEXP.** Figure 4 gives some insight into the dynamics of ALEXP when $M = 165$. In particular, it shows how ALEXP can rule out sub-optimal agents without ever having queried them. Figure (a) shows the distribution $q_t$, at $t = 20$ which is roughly equal to the optimal $n_0$ for ETC in this configuration. The oracle model $j^\star$ is annotated with a star, and has the highest probability of selection. We observe that, already at this time step, more than $80\%$ of the agents are practically ruled out, due to small probability of selection. However, according to Figure (b), which shows $M_t$ the total number of visited models, less than $10\%$ of the models are queried at $t = 20$. This is the key practical benefit of ALEXP compared to black-box algorithms such as CORRAL. Lastly, Figure (c) shows how $q_{t,j^\star}$ the probability of selecting the oracle agent changes with time. While this probability is higher than that of the other agents, Figure (c) shows that $q_{t,j^\star}$ is not exceeding $0.25$, therefore there is always a probability of greater than $0.75$ that we sample another agent, making ALEXP robust to hard problem instances where many agents perform efficiently. We conclude that ALEXP seems to rapidly recognize the better performing agents, and select among them with high probability.

## 7 Conclusion

We proposed ALEXP, an algorithm for simultaneous online model selection and bandit optimization. As a first, our approach leads to anytime valid guarantees for model selection and bandit regret, and does not rely on a priori determined exploration schedule. Further, we showed how the Lasso can be used together with the exponential weights algorithm to construct a low-variance online learner. This new connection between high-dimensional statistics and online learning opens up avenues for future research on high-dimensional online learning. We established empirically that ALEXP has favorable exploration–exploitation dynamics, and outperforms existing baselines. We tackled the open problem of Agarwal et al. [2017], and showed that $\log M$ dependency for regret is achievable for linearly parametrizable rewards. This problem remains open for more general, non-parametric reward classes.

## Acknowledgments

We thank Jonas Rothfuss and Miles Wang-Henderson for their valuable suggestions regarding the writing. We thank Scott Sussex and Felix Schur for their thorough feedback. This research was supported by the European Research Council (ERC) under the European Union's Horizon 2020 research and Innovation Program Grant agreement no. 815943. Nicolas Emmenegger was supported by the Zeno Karl Schindler Foundation and the Swiss Study Foundation. Aldo Pacchiano would like to thank the support of the Eric and Wendy Schmidt Center at the Broad Institute of MIT and Harvard. This work was supported in part by funding from the Eric and Wendy Schmidt Center at the Broad Institute of MIT and Harvard.

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

# Contents of Appendix

## A  Extended Literature Review

The sparse linear bandit literature considers linear reward functions of the form $\boldsymbol{\theta}^\top \boldsymbol{x}$, where $\boldsymbol{x} \in \mathbb{R}^p$, however a sub-vector of size $d$ is sufficient to span the reward function. This can be formulated as model selection among $M = \binom{p}{d}$ different linear parametrizations, where each $\boldsymbol{\phi}_j$ is a $d$-dimensional feature map. We present the bounds in terms of $d$ and $M$ for coherence with the rest of the text, assuming that $M = \mathcal{O}(p)$, which is the case when $d \ll p$.

Table 2 compares recent work on sparse linear bandits based on a number of important factors. In this table, the ETC algorithms follow the general format of exploring, performing parameter estimation once at $t = n_0$, and then repeatedly suggesting the same action which maximizes $\hat{\boldsymbol{\theta}}_{n_0}^\top \boldsymbol{\phi}(\boldsymbol{x})$. Explore-then-($\epsilon$)Greedy takes a similar approach, however it does not settle on $\hat{\boldsymbol{\theta}}_{n_0}$, rather it continues to update the parameter estimate and select $\boldsymbol{x}_t = \arg\max \hat{\boldsymbol{\theta}}_t^\top \boldsymbol{\phi}(\boldsymbol{x})$. The UCB algorithms iteratively update upper confidence bound, and choose actions which maximize them. The regret bounds in Table 2 are simplified to the terms with largest rate of growth, *the reader should check the corresponding papers for rigorous results*. Some of the mentioned bounds depend on problem-dependent parameters (e.g. $c_K$), which may not be treated as absolute constants and have complicated forms. To indicate such parameters we use $\tau$ in Table 2, following the notation of Hao et al. [2020]. Note that $\tau$ varies across the rows of the table, and is just an indicator for existence of other terms.

Abbasi-Yadkori et al. [2012] use the SEQSEW online regression oracle [Gerchinovitz, 2011] for estimating the parameter vector, together with a UCB policy. The regression oracle is an exponential weights algorithm, which runs on the squared error loss. This subroutine, and thereby the algorithm proposed by Abbasi-Yadkori et al. [2012] are not computationally efficient, and this is believed to be unavoidable. This work considers the data-rich regime and shows $R(n) = \mathcal{O}(\sqrt{dMn})$, matching the lower bound of Theorem 24.3 in Lattimore and Szepesvári [2020].

Carpentier and Munos [2012] assume that the action set is a Euclidean ball, and that the noise is directly added to the parameter vector, i.e. $y_t = \boldsymbol{x}^\top (\boldsymbol{\theta} + \boldsymbol{\varepsilon}_t)$. Roughly put, linear bandits with parameter noise are "easier" to solve than stochastic linear bandits with reward noise, since the noise is scaled proportionally to the features $x_i$ and does less "damage " [Chapter 29.3 Lattimore and Szepesvári, 2020]. In this setting, Carpentier and Munos [2012] present a $\mathcal{O}(d\sqrt{n})$ regret bound.

Recent work considers contextual linear bandits, where at every step $\mathcal{A}_t$, a stochastic finite subset of size $K$ from $\mathcal{A}$, is presented to the agent. It is commonly assumed that members of $\mathcal{A}_t$ are i.i.d., and the sampling distribution is diverse and time-independent. The diversity assumption is often in the

form of a restricted eigenvalue condition (Definition 2) on the covariance of the context distribution [e.g. in, Kim and Paik, 2019, Bastani and Bayati, 2020]. Li et al. [2022] require a stronger condition which directly assumes that $\lambda_{\min}(\Phi_t)$ the minimum eigenvalue of the empirical covariance matrix is lower bounded. This is generally not true, but may hold with high probability. Hao et al. [2020] assume that the action set spans $\mathbb{R}^{dM}$. We believe that this assumption is the weakest in the literature, and conjecture that it is necessary for model selection. If not met, the agent can not explore in all relevant directions, and may not identify the relevant features. Our diversity assumption is similar to Hao et al. [2020], adapted to our problem setting. Mainly, we consider reward functions which are linearly parametrizable, i.e. $\boldsymbol{\theta}^\top \boldsymbol{\phi}(\boldsymbol{x})$, as oppose to linear rewards, i.e. $\boldsymbol{\theta}^\top \boldsymbol{x}$.

A key distinguishing factor between ALEXP and existing work on sparse linear bandit is that ALEXP is horizon-independent and does not rely on a forced exploration schedule. As shown on Table 2, majority of prior work relies either on an initial exploration stage, the length of which is determined according to $n$ [e.g., Carpentier and Munos, 2012, Kim and Paik, 2019, Li et al., 2022, Hao et al., 2020, Jang et al., 2022], or on a hand crafted schedule, which is again designed for a specific horizon [Bastani and Bayati, 2020]. Oh et al. [2021], which analyzes $K$-armed contextual bandits, does not require explicit exploration, and instead imposes restrictive assumptions on the diversity of context distribution, e.g. relaxed symmetry and balanced covariance. Regardless, the regret bounds hold in expectation, and are not time-uniform.

Table 2: Overview of recent work on high-dimensional Bandits. Parameter $\tau$ shows existence of other problem-dependent terms which are not constants, and varies across different rows. The regret bounds are simplified and are *not* rigorous.

| | $|\mathcal{A}_t|$ | data-poor | adap. exp. | any-time | action selection policy | MS algo | context or action assump. | Regret |
|---|---|---|---|---|---|---|---|---|
| Abbasi-Yadkori et al. | $\infty$ | ✗ | ✓ | ✓ | UCB | EXP4 on Sqrd error | $\mathcal{A}$ is compact | $\sqrt{dMn}$, w.h.p. |
| Foster et al. | K | ✓ | ✓ | ✗ | UCB | EXP4 on Sqrd error | $\lambda_{\min}(\Sigma) \geq c_\lambda$ | $(Mn)^{3/4}K^{1/4} + \sqrt{KdMn}$ w.h.p |
| Carpentier and Munos | $\infty$ | ✓ | ✗ | ✗ | UCB | Hard Thresh. | $\mathcal{A}$ is a ball param. noise | $d\sqrt{n}$, w.h.p. |
| Bastani and Bayati | K | ✗ | ✗ | ✗ | Explore then Greedy | Lasso | $\kappa(\Sigma) > c_K$ | $\tau K d^2 (\log n + \log M)^2$ w.h.p. |
| Kim and Paik | K | ✗ | ✗ | ✗ | Explore then $\epsilon$-Greedy | Lasso | $\kappa(\Sigma) > c_K$ | $\tau d\sqrt{n}\log(Mn)$, w.h.p. |
| Oh et al. | K | ✓ | ✓ | ✗ | Greedy | Lasso | $\kappa(\Sigma) > c_\kappa$ + other assums. | $\tau d\sqrt{n}\log(Mn)$ in expectation |
| Li et al. | K | ✓ | ✗ | ✗ | ETC | Lasso | $\lambda_{\min}(\hat{\Sigma}) > c_\lambda$ | $\tau(n^2 d)^{1/3}\sqrt{\log Mn}$ in expectation |
| Hao et al. | $\infty$ | ✓ | ✗ | ✗ | ETC | Lasso | $\mathcal{A}$ spans $\mathbb{R}^{dM}$ + is compact | $(ndC_{\min}^{-1})^{2/3}(\log M)^{1/3}$ w.h.p. |
| Jang et al. | $\infty$ | ✓ | ✗ | ✗ | ETC | Hard Thresh. | $\mathcal{A} \subset [-1,1]^{Md}$ + spans $\mathbb{R}^{Md}$ | $(nd)^{2/3}(C_{\min}^{-1}\log M)^{1/3}$ w.h.p. |
| ALEXP (Ours) | $\infty$ | ✓ | ✓ | ✓ | Greedy or UCB | EXP4 on reward est. | $\text{Im}(\phi_j)$ spans $\mathbb{R}^d$ $\mathcal{A}$ is compact | $\sqrt{n\log M}(n^{1/4} + C_{\min}^{-1}\log M)$ w.h.p |

# B  Time Uniform Lasso Analysis

We start by showing that the sum of squared sub-gaussian variables is a sub-Gamma process (c.f. Definition 22).

**Lemma 4** (Empirical Process is sub-Gamma). *For $t \geq 1$, suppose $\xi_t$ are a sequence conditionally standard sub-Gaussians adapted to the filtration $\mathcal{F}_t = \sigma(\xi_1, \ldots, \xi_t)$. Let $v_t \in \mathbb{R}$, and $Z_t := \xi_t^2 - 1$. Define the processes $S_t := \sum_{i=1}^t Z_i v_i$ and $V_t := 4\sum_{i=1}^t v_i^2$. Then $(S_t)_{t=0}^\infty$ is sub-Gamma with variance process $(V_t)_{t=0}^\infty$ and scale parameter $c = 4\max_{i\geq 1} v_i$.*

*Proof of Lemma 4.* By definition [c.f. Definition 1, Howard et al., 2021], $S_t$ is sub-Gamma if for each $\lambda \in [0, 1/c)$, there exists a supermartingale $(M_t(\lambda))_{t=0}^\infty$ w.r.t. $\mathcal{F}_t$, such that $\mathbb{E}\, M_0 = 1$ and for all $t \geq 1$:

$$\exp\left\{\lambda S_t - \frac{\lambda^2}{2(1-c\lambda)}V_t\right\} \leq M_t(\lambda) \qquad a.s.$$

We show the above holds in equality by proving that the left hand side itself, is a supermartingale w.r.t. $\mathcal{F}_t$. We define, $M_t(\lambda) := \exp\{\lambda S_t - \lambda^2 V_t/2(1 - c\lambda)\}$, therefore,

$$\mathbb{E}\left[M_t|\mathcal{F}_{t-1}\right] \leq \mathbb{E}\left[\exp\left(\lambda S_{t-1} - \frac{\lambda^2}{2(1-c\lambda)}V_{t-1} + \lambda Z_t v_t - \frac{2\lambda^2 v_t^2}{(1-c\lambda)}\right)|\mathcal{F}_{t-1}\right]$$

$$= \mathbb{E}\left[M_{t-1}|\mathcal{F}_{t-1}\right]\mathbb{E}\left[\exp\left(\lambda Z_t v_t\right)|\mathcal{F}_{t-1}\right]\exp\left(-\frac{2\lambda^2 v_t^2}{1-c\lambda}\right)$$

$$= M_{t-1}\mathbb{E}\left[\exp\left(\lambda Z_t v_t\right)|\mathcal{F}_{t-1}\right]\exp\left(-\frac{2\lambda^2 v_t^2}{1-c\lambda}\right).$$

Note that $Z_t$ is $\mathcal{F}_{t-1}$-measurable, conditionally centered and conditionally sub-exponential with parameters $(\nu, \alpha) = (2, 4)$ (c.f. Vershynin [2018, Lemma 2.7.6] and Wainwright [2019, Example 2.8]). Therefore, for $\lambda < 1/c$,

$$\mathbb{E}\left[\exp\left(\lambda v_t Z_t\right)|\mathcal{F}_{t-1}\right] \leq \exp\left(2\lambda^2 v_t^2\right) \leq \exp\left(\frac{2\lambda^2 v_t^2}{1-c\lambda}\right),$$

where the last inequality holds due to the fact that $0 \leq 1 - c\lambda < 1$. Therefore,

$$\mathbb{E}\left[M_t|\mathcal{F}_{t-1}\right] \leq M_{t-1}\exp\left(\frac{2\lambda^2 v_t^2}{1-c\lambda}\right)\exp\left(-\frac{2\lambda^2 v_t^2}{1-c\lambda}\right) = M_{t-1}.$$

for $\lambda \in [0, 1/c)$, concluding the proof. $\qquad\square$

We now construct a self-normalizing martingale sequence based on $\ell_2$-norm of the empirical process error term, and recognize that it is a sub-gamma process. We then employ our curved Bernstein bound Lemma 25 to control the boundary. This step will allow us to "ignore" the empirical process error term later in the lasso analysis.

**Lemma 5** (Empirical Process is dominated by regularization.). *Let*

$$A_j = \left\{\forall t \geq 1:\ \left\|(\Phi_t^\top \boldsymbol{\varepsilon}_t)_j\right\|_2/t \leq \lambda_t/2\right\}.$$

*Then, for any $0 \leq \delta < 1$, the event $A = \cap_{j=1}^M A_j$ happens with probability $1 - \delta$, if for all $t \geq 1$,*

$$\lambda_t \geq \frac{2\sigma}{\sqrt{t}}\sqrt{1 + \frac{5}{\sqrt{2}}\sqrt{d\left(\log(2M/\delta) + (\log\log d)_+\right)} + \frac{12}{\sqrt{2}}\left(\log(2M/\delta) + (\log\log d)_+\right)}.$$

*Proof of Lemma 5.* This proof includes a treatment of the empirical process similar to Lemma B.1 in Kassraie et al. [2022], but adapts it to our time-uniform setting. Since $\varepsilon_i$ are zero-mean sub-gaussian variables, as driven in Lemma 3.1 [Lounici et al., 2011], it holds that

$$A_j^c = \left\{\exists t:\ \frac{1}{t^2}\boldsymbol{\varepsilon}_t^T \Phi_{t,j}\Phi_{t,j}^\top \boldsymbol{\varepsilon}_t \geq \frac{\lambda^2}{4}\right\} = \left\{\exists t:\ \frac{\sum_{i=1}^t v_i(\xi_i^2 - 1)}{\sqrt{2}\|\boldsymbol{v}_t\|} \geq \alpha_{t,j}\right\}$$

where $\xi_i$ are sub-gaussian variables with variance proxy 1, scalar $v_i$ denotes the $i$-th eigenvalue of $\Phi_{t,j}\Phi_{t,j}^\top/t$ with the concatenated vector $\boldsymbol{v}_t = (v_1, \ldots, v_t)$, and

$$\alpha_{t,j} = \frac{t^2\lambda^2/(4\sigma^2) - \mathrm{tr}(\Phi_{t,j}^\top \Phi_{t,j})}{\sqrt{2}\left\|\Phi_{t,j}^\top \Phi_{t,j}\right\|_{\mathrm{Fr}}}.$$

We can apply Lemma 25 to control the probability of event $A_j^c$ by tuning $\lambda$. Mainly, for $A_j^c$ to happen with probability less than $\delta/M$, Lemma 25 states that the following must hold for all $t$,

$$\sqrt{2}\|\boldsymbol{v}_t\|_2\alpha_{t,j} \geq \frac{5}{2}\sqrt{\max\left\{4\|\boldsymbol{v}_t\|_2^2, 1\right\}\omega_{\delta/M}(\|\boldsymbol{v}_t\|_2)} + 12\omega_{\delta/M}(\|\boldsymbol{v}_t\|_2)\max_{t\geq 1} v_t \qquad (4)$$

Recall that w.l.o.g. feature maps are bounded everywhere $\|\phi_j(\cdot)\|_2 \leq 1$, and $\mathrm{rank}(\Phi_j) \leq d$ which allows for the following matrix inequalities,

$$\mathrm{tr}(\Phi_{t,j}^\top \Phi_{t,j}) = \mathrm{tr}(\Phi_{t,j}\Phi_{t,j}^\top) = \sum_{i=1}^t \phi_j^\top(x_i)\phi_j(x_i) \leq t$$

$$\left\|\Phi_{t,j}\Phi_{t,j}^\top\right\| \leq \operatorname{tr}(\Phi_{t,j}\Phi_{t,j}^\top) \leq t$$
$$\left\|\Phi_{t,j}\Phi_{t,j}^\top\right\| \leq \left\|\Phi_{t,j}\Phi_{t,j}^\top\right\|_{\mathrm{Fr}} \leq \sqrt{d}\left\|\Phi_{t,j}\Phi_{t,j}^\top\right\| \leq t\sqrt{d}$$

Therefore,
$$\|\boldsymbol{v}_t\| = \left\|\Phi_{t,j}\Phi_{t,j}^\top\right\|_{\mathrm{Fr}}/t \leq \sqrt{d}, \quad \max_{t\geq 1} v_t = \max_{t\geq 1}\left\|\Phi_{t,j}\Phi_{t,j}^\top\right\|/t \leq 1.$$

For Eq. (4) to hold, is suffices that for all $t \geq 1$,

$$\lambda \geq \frac{2\sigma}{\sqrt{t}}\sqrt{1 + \frac{5}{2\sqrt{2}}\sqrt{4d\left(\log(2M/\delta) + (\log\log d)_+\right)} + \frac{12}{\sqrt{2}}\left(\log(2M/\delta) + (\log\log d)_+\right)}.$$

Therefore, if $\lambda_t$ are chosen to satisfy the above inequality, each $A_j^c$ happens with probability less than $\delta/M$. Then by applying union bound, $\cup_{j=1}^M A_j^c$ happens with probability less than $\delta$. $\qquad\square$

***Proof of Theorem 3.*** The theorem statement requires that the regularization parameter $\lambda_t$ is chosen such that condition of Lemma 5 is met, and therefore event $A$ happens with probability $1 - \delta$. Throughout this proof, which adapts the analysis of Theorem 3.1. Lounici et al. [2011] to the time-uniform setting, we condition on $A$ happening, and later incorporate the probability.

**Step 1.** Let $\hat{\boldsymbol{\theta}}_t$ be a minimizer of $\mathcal{L}$ and $\boldsymbol{\theta}$ be the true coefficients vector, then $\mathcal{L}(\hat{\boldsymbol{\theta}}_t; H_t, \lambda_t) \leq \mathcal{L}(\boldsymbol{\theta}; H_t, \lambda_t)$. Writing out the loss and re-ordering the inequality we obtain,

$$\frac{1}{t}\left\|\Phi_t(\hat{\boldsymbol{\theta}}_t - \boldsymbol{\theta})\right\|_2^2 \leq \frac{2}{t}\boldsymbol{\varepsilon}_t^T\Phi_t(\hat{\boldsymbol{\theta}}_t - \boldsymbol{\theta}) + 2\lambda_t\sum_{j=1}^M\left(\|\boldsymbol{\theta}_j\|_2 - \left\|\hat{\boldsymbol{\theta}}_{t,j}\right\|_2\right).$$

which is often referred to as the Basic inequality [Bühlmann and Van De Geer, 2011]. By Cauchy-Schwarz and conditioned on event $A$,

$$\boldsymbol{\varepsilon}_t^T\Phi_t(\hat{\boldsymbol{\theta}}_t - \boldsymbol{\theta}) \leq \sum_{j=1}^M\left\|(\Phi_t^T\boldsymbol{\varepsilon}_t)_j\right\|_2\left\|\hat{\boldsymbol{\theta}}_{t,j} - \boldsymbol{\theta}_j\right\|_2 \leq \frac{t\lambda}{2}\sum_{j=1}^M\left\|\hat{\boldsymbol{\theta}}_{t,j} - \boldsymbol{\theta}_j\right\|_2$$

then adding $\lambda_t\sum_{j=1}^M\left\|\hat{\boldsymbol{\theta}}_{t,j} - \boldsymbol{\theta}_j\right\|_2$ to both sides, applying the triangle inequality, and recalling from Section 3 that $\boldsymbol{\theta}_j = 0$ for $j \neq j^\star$ gives

$$\frac{1}{t}\left\|\Phi_t(\hat{\boldsymbol{\theta}}_t - \boldsymbol{\theta})\right\|_2^2 + \lambda_t\sum_{j=1}^M\left\|\hat{\boldsymbol{\theta}}_{t,j} - \boldsymbol{\theta}_j\right\|_2 \leq 2\lambda_t\sum_{j=1}^M\left\|\hat{\boldsymbol{\theta}}_{t,j} - \boldsymbol{\theta}_j\right\|_2 + 2\lambda_t\sum_{j=1}^M\left(\|\boldsymbol{\theta}_j\|_2 - \left\|\hat{\boldsymbol{\theta}}_{t,j}\right\|_2\right)$$
$$\leq 4\lambda_t\left\|\hat{\boldsymbol{\theta}}_{t,j^\star} - \boldsymbol{\theta}_{j^\star}\right\|_2.$$

Since each term on the left hand side is positive, then each is also individually smaller than the right hand side, and we obtain,

$$\frac{1}{t}\left\|\Phi_t(\hat{\boldsymbol{\theta}}_t - \boldsymbol{\theta})\right\|_2^2 \leq 4\lambda_t\left\|\hat{\boldsymbol{\theta}}_{t,j^\star} - \boldsymbol{\theta}_{j^\star}\right\|_2 \tag{5}$$

$$\sum_{\substack{j=1 \\ j\neq j^\star}}^M\left\|\hat{\boldsymbol{\theta}}_{t,j} - \boldsymbol{\theta}_j\right\|_2 \leq 3\left\|\hat{\boldsymbol{\theta}}_{t,j^\star} - \boldsymbol{\theta}_{j^\star}\right\|_2 \tag{6}$$

**Step 2.** Consider a sequence $(c_1, \ldots, c_k, \ldots)$, where $c_1 \geq \cdots \geq c_k \ldots$, then

$$c_k \leq \frac{1}{k}(kc_k + \sum_{i>k} c_i) \leq \sum_{i\geq 1}\frac{c_i}{k}. \tag{7}$$

Define $J_1 = \{j^\star\}$ and $J_2 = \{j^\star, j'\}$ where

$$j' = \operatorname*{arg\,max}_{\substack{j\in[M] \\ j\neq j^\star}}\left\|\hat{\boldsymbol{\theta}}_{t,j} - \boldsymbol{\theta}_j\right\|_2.$$

For any $J \subset [M]$ the complementing set is denoted as $J^c = [M] \setminus J$. For simplicity let $c_j = \left\| \hat{\boldsymbol{\theta}}_{t,j} - \boldsymbol{\theta}_j \right\|_2$, and let $\pi(k)$ denote the index of the $k$-th largest element of $\{c_j : j \in J_1^c\}$. By definition of $J_2^c$ we have,

$$\sum_{j \in J_2^c} \left\| \hat{\boldsymbol{\theta}}_{t,j} - \boldsymbol{\theta}_j \right\|_2^2 = \sum_{\substack{k>1 \\ \pi(k) \in J_1^c}} c_k^2 \overset{(7)}{\le} \sum_{\substack{k>1 \\ \pi(k) \in J_1^c}} \frac{(\sum_{i \in J_1^c} c_i)^2}{k^2}$$

$$\le \big( \sum_{i \in J_1^c} c_i \big)^2 \sum_{\substack{k>1 \\ \pi(k) \in J_1^c}} \frac{1}{k^2} \overset{(6)}{\le} 9 c_{j^\star}^2$$

$$\le 9(c_{j^\star}^2 + c_{j'}^2) = 9 \sum_{j \in J_2} \left\| \hat{\boldsymbol{\theta}}_{t,j} - \boldsymbol{\theta}_j \right\|_2^2,$$

which, in turn, gives

$$\left\| \hat{\boldsymbol{\theta}}_t - \boldsymbol{\theta} \right\|_2 = \sqrt{\sum_{j=1}^M \left\| \hat{\boldsymbol{\theta}}_{t,j} - \boldsymbol{\theta}_j \right\|_2^2} \le \sqrt{10 \sum_{j \in J_2} \left\| \hat{\boldsymbol{\theta}}_{t,j} - \boldsymbol{\theta}_j \right\|_2^2}. \tag{8}$$

**Step 3.** On the other hand, due to (6), and by definition of $J_2$ it also holds that

$$\sum_{j \in J_2^c} \left\| \hat{\boldsymbol{\theta}}_{t,j} - \boldsymbol{\theta}_j \right\|_2 \le 3 \sum_{j \in J_2} \left\| \hat{\boldsymbol{\theta}}_{t,j} - \boldsymbol{\theta}_j \right\|_2.$$

From the theorem assumptions and Definition 2, we know that there exists $0 < \kappa(\Phi_t, 2)$, therefore by Definition 2, the feature matrix $\Phi_t$ satisfies,

$$\sum_{j \in J_2} \left\| \hat{\boldsymbol{\theta}}_{t,j} - \boldsymbol{\theta}_j \right\|_2^2 \le \frac{1}{t \kappa^2(\Phi_t, 2)} \left\| \Phi_t(\hat{\boldsymbol{\theta}}_t - \boldsymbol{\theta}) \right\|_2^2$$

$$\overset{(5)}{\le} \frac{1}{\kappa^2(\Phi_t, 2)} 4 \lambda_t \left\| \hat{\boldsymbol{\theta}}_{t,j^\star} - \boldsymbol{\theta}_{j^\star} \right\|_2 \le \frac{1}{\kappa^2(\Phi_t, 2)} 4 \lambda_t \sqrt{\sum_{j \in J_2} \left\| \hat{\boldsymbol{\theta}}_{t,j} - \boldsymbol{\theta}_j \right\|_2^2}.$$

From here, by applying (8) we get,

$$\left\| \hat{\boldsymbol{\theta}}_t - \boldsymbol{\theta} \right\|_2 \le \frac{4\sqrt{10} \lambda_t}{\kappa^2(\Phi_t, 2)}.$$

If $\lambda_t$ are chosen according to Lemma 5, event $A$ and, in turn, the inequality above hold with probability greater than $1 - \delta$. $\qquad \square$

## C  Results on Exploration

In this section we present lower-bounds on the eigenvalues of the covariance matrix $\Phi_t \Phi_t^\top$, as it is later used in our regret analysis. In particular, we show that the feature matrix $\Phi_t$ satisfies the restricted eigenvalue condition (Definition 2) required for valid Lasso confidence set (Theorem 3), and calculate a lower bound on $\kappa(\Phi_t, 2)$. The lower bound is later used by Lemma 19 and Lemma 20 to develop the model selection regret. We show this bound in three steps.

Equivalent to Definition 2, we write $\kappa(\Phi_t, s) = \inf_{b \in \Xi_s} \| \Phi_t b \|_2 / \sqrt{t}$ where

$$\Xi_s := \left\{ b \in \mathbb{R}^d \setminus \{0\} \Big| \sum_{j \notin J} \| b_j \|_2 \le 3 \sum_{j \in J} \| b_j \|_2, \sqrt{\sum_{j \in J} \| b_j \|_2^2} \le 1 \text{ s.t. } J \subset \{1, \dots, M\}, |J| \le s. \right\}. \tag{9}$$

For simplicity in notation, we further define

$$\tilde{\kappa}(A, s) := \min_{b \in \Xi_s} b^\top A b. \tag{10}$$

since $\tilde{\kappa}(\frac{\Phi_t^\top \Phi_t}{t}, s) = \kappa^2(\Phi_t, s)$.

**Step I.** Consider the exploratory steps at which $\alpha_t = 1$. Let $\Phi_{\pi,t}$ be a sub-matrix of $\Phi_t$ where only rows from exploratory steps are included. Note that $\Phi_{\pi,t} \in \mathbb{R}^{t' \times dM}$ is a random matrix, where the number of rows $t'$ are also random. We show that $\kappa^2(\Phi_t, s)$ is lower bounded by $\kappa^2(\Phi_{t,\pi}, s)$.

**Lemma 6.** *Suppose $\Phi_{\pi,t}$ has $t'$ rows. Then,*

$$\kappa^2(\Phi_t, s) \geq \frac{t'}{t} \kappa^2(\Phi_{\pi,t}, s)$$

*Proof of Lemma 6.* Let $\Psi^{(t)} = \phi(\boldsymbol{x}_t)\phi^\top(\boldsymbol{x}_t) \in \mathbb{R}^{dM \times dM}$ for all $t = 1, \ldots, n$. Note that $\Psi^{(t)}$ is positive semi-definite by construction. We have,

$$\|\Phi_t b\|_2^2 = \sum_{s=1}^t b^\top \Psi^{(s)} b = \sum_{s \in T_\pi} b^\top \Psi^{(s)} b + \sum_{s \notin T_\pi} b^\top \Psi^{(s)} b$$

where the set $T_\pi$ contains the indices of the exploratory steps at which the action is selected according to $\pi$. Therefore,

$$\kappa^2(\Phi_t, s) = \frac{1}{t} \min_{b \in \Xi_s} \|\Phi_t \boldsymbol{b}\|_2^2$$

$$= \min_{b \in \Xi_s} \frac{1}{t} \sum_{s \in T_\pi} b^\top \Psi^{(s)} b + \frac{1}{t} \sum_{s \notin T_\pi} b^\top \Psi^{(s)} b$$

$$\geq \min_{b \in \Xi_s} \frac{1}{t} \sum_{s \in T_\pi} b^\top \Psi^{(s)} b$$

where the last inequality holds due to $\Psi^{(s)}$ being PSD. Then we have,

$$\kappa^2(\Phi_t, s) \geq \min_{b \in \Xi_s} b^\top \left( \frac{1}{t} \sum_{s \in T_\pi} \Psi^{(s)} \right) b = \frac{|T_\pi|}{t} \kappa^2(\Phi_{\pi,t}, s)$$

$\square$

While the number of rows of $\Phi_{\pi,t}$ is a random variable, we continue to condition on the event that $\Phi_{\pi,t}$ has $t'$ rows, and investigate the distribution of its restricted eigenvalues.

**Step II.** The restricted eigenvalues of the *exploratory* submatrix are well bounded away from zero.

**Lemma 7.** *Let $\pi$ be the solution to* (2)*, and $s \in \mathbb{N}$. Suppose $\Phi_{\pi,t}$ has $t'$ rows. Then for all $\delta > 0$,*

$$\mathbb{P}\left( \forall t' : \kappa^2(\Phi_{\pi,t}, s) \geq \tilde{\kappa}(\Sigma, s) - \frac{80s}{\sqrt{t'}} \sqrt{\log(2Md/\delta) + (\log\log 4t')_+} \right) \geq 1 - \delta$$

*where $\Sigma = \Sigma(\pi, \phi) := \mathbb{E}_{\boldsymbol{x} \sim \pi} \phi(\boldsymbol{x})\phi^\top(\boldsymbol{x})$ and $\tilde{\kappa}$ is defined in* (10)*.*

**Step III.** Remains to combine the two above lemmas and incorporate a high probability bound on $t'$, showing that it is close to $\sum_{s=1}^t \gamma_s$.

**Lemma 8.** *There exist absolute constants $C_1, C_2$ which satisfy,*

$$\mathbb{P}\left( \forall t \geq 1 : \kappa^2(\Phi_t, 2) \geq C_1 \tilde{\kappa}(\Sigma, 2) t^{-1/4} - C_2 t^{-5/8} \sqrt{\log(Md/\delta) + (\log\log t)_+} \right) \geq 1 - \delta$$

*if $\gamma_t = \mathcal{O}(t^{-1/4})$. Let $\pi$ be the solution to* (2)*, then it further holds that*

$$\mathbb{P}\left( \forall t \geq 1 : \kappa^2(\Phi_t, 2) \geq C_1 C_{\min} t^{-1/4} - C_2 t^{-5/8} \sqrt{\log(Md/\delta) + (\log\log t)_+} \right) \geq 1 - \delta$$

The regret analysis of Hao et al. [2020] also relies on connecting $\kappa(\Phi_t, s)$ to $C_{\min}$, and for this, they use Theorem 2.4 of Javanmard and Montanari [2014]. This theorem states that there exists a problem-dependent constant $C_1$ for which $\kappa^2(\Phi_t, s) \geq C_1 C_{\min}$ with high probability, if $t \geq n_0$ and roughly $n_0 = \mathcal{O}(\sqrt[3]{n^2 \log M})$. We highlight that Lemma 8, presents a lower-bound which holds for all $t \geq 1$, however this comes at the cost of getting a looser lower bound than the result of Javanmard and Montanari [2014] for the larger time steps $t$. In fact, due to the sub-optimal dependency of Lemma 8 on $t$, we later obtain sub-optimal dependency on the horizon for the case when $n \gg M$. It is unclear to us if this rate can be improved without assuming knowledge of $n$, or that $n \geq n_0$.

For the last lemma in this section we show that the empirical sub-matrices $\Phi_{t,j}$ are also bounded away from zero. This will be required later to prove Lemma 15.

**Lemma 9** (Base Model $\lambda_{\min}$ Bound). *Assume $\pi$ is the maximizer of Eq. (2). Then, with probability greater than $1 - \delta$, simultaneously for all $j = 1, \ldots, M$ and $t \geq 1$,*

$$\lambda_{\min}(\Phi_{t,j}^{\top}\Phi_{t,j}) \geq C_1 C_{\min} t^{3/4} - C_2 t^{3/8}\sqrt{\log(Md/\delta) + (\log\log t)_+}$$

*if $\gamma_t = \mathcal{O}(t^{-1/4})$.*

## C.1  ALEXP with Uniform Exploration

We presented our main regret bound (Theorem 1) in terms of $C_{\min}$, which only depends on properties of the feature maps and the action domain. We give a lower-bound on $C_{\min}$ for a toy scenario which corresponds to the problem of linear feature selection over convex action sets.

**Proposition 10** (Invertible Features). *Suppose $\phi(x) := Ax : \mathbb{R}^d \to \mathbb{R}^d$ is an invertible linear map, and $\mathcal{X} \in \mathbb{R}^d$ is a convex body. Then,*

$$C_{\min} \geq \frac{\lambda_{\min}(A)}{\lambda_{\max}^2(T)} > 0$$

*where $T$ is the transformation which maps $\mathcal{X}$ to an isotropic body.*

The lower-bound of Proposition 10 is achieved by simply exploring via $\pi = \text{Unif}(\mathcal{X})$. Inspired by Schur et al. [2023, Lemma E.13], we show that even for non-convex action domains and orthogonal feature maps, the uniform exploration yields a constant lower-bound on restricted eigenvalues.

**Proposition 11** (Orthonormal Features). *Suppose $\phi_j : \mathcal{X} \to \mathbb{R}$ are chosen from an orthogonal basis of $L^2(\mathcal{X})$, and satisfy $\|\phi_i\|_{L_\mu^2(\mathcal{X})}/Vol(\mathcal{X}) \geq 1$. Then there exist absolute constants $C_1$ and $C_2$ for which the exploration distribution $\pi = \text{Unif}(\mathcal{X})$ satisfies*

$$\mathbb{P}\left(\forall t \geq 1 : \ \kappa^2(\Phi_t, 2) \geq C_1 t^{-1/4} - C_2 t^{-5/8}\sqrt{\log(Md/\delta) + (\log\log t)_+}\right) \geq 1 - \delta.$$

The $d = 1$ condition is met without loss of generality, by splitting the higher dimensional feature maps and introducing more base features, which will increase $M$. Moreover, the orthonormality condition is met by orthogonalizing and re-scaling the feature maps. Basis functions such as Legendre polynomials and Fourier features [Rahimi et al., 2007] satisfy these conditions.

By invoking Proposition 11, instead of Lemma 8 in the proof of Theorem 1, we obtain the regret of ALEXP with uniform exploration.

**Corollary 12** (ALEXP with Uniform Exploration). *Let $\delta \in (0, 1]$. Suppose $\phi_j : \mathcal{X} \to \mathbb{R}$ are chosen from an orthogonal basis of $L^2(\mathcal{X})$, and satisfy $\|\phi_i\|_{L_\mu^2(\mathcal{X})}/Vol(\mathcal{X}) \geq 1$. Assume the oracle agent employs a UCB or a Greedy policy, as laid out in Section 5. Choose $\eta_t = \mathcal{O}(1/\sqrt{t}C(M, \delta, d))$ and $\gamma_t = \mathcal{O}(t^{-1/4})$ and $\lambda_t = \mathcal{O}(C(M, \delta, d)/\sqrt{t})$, then ALEXP with uniform exploration $\pi = \text{Unif}(\mathcal{X})$ attains the regret*

$$R(n) = \mathcal{O}\Big(Bn^{3/4} + \sqrt{n}C(M, \delta, d)\log M + B^2\sqrt{n} + B\sqrt{n\left((\log\log nB^2)_+ + \log(1/\delta)\right)}$$

$$+ (n^{3/4} + \log n)C(M, \delta, d) + n^{5/8}\sqrt{d\log n + \log(1/\delta) + B^2}\Big)$$

*with probability greater than $1 - \delta$, simultaneously for all $n \geq 1$. Here,*

$$C(M, \delta, d) = \mathcal{O}\left(\sqrt{1 + \sqrt{d\left(\log(M/\delta) + (\log\log d)_+\right)}} + (\log(M/\delta) + (\log\log d)_+)\right).$$

## C.2  Proof of Results on Exploration

As an intermediate step, we consider the restricted eigenvalue property of the empirical covariance matrix. Given $t'$ samples, the empirical estimate of $\Sigma$ is

$$\hat{\Sigma}_{t'} := \frac{1}{t'}\sum_{s=1}^{t'}\phi(x_s)\phi^{\top}(x_s) \tag{11}$$

where $x_s$ are sampled according to $\pi$. We show that every entry of $\hat{\Sigma}_{t'}$ is close to the corresponding entry in $\Sigma$, and later use it in the proofs of eigenvalue lemmas.

**Lemma 13** (Anytime Bound for The Entries of Empirical Covariance Matrix). *Let $\hat{\Sigma}_{t'}$ be the empirical covariance matrix corresponding to $\Sigma(\pi, \phi)$ given $t'$ samples. Then,*

$$\mathbb{P}\left(\exists t' : \; d_\infty(\Sigma, \hat{\Sigma}_{t'}) \geq \frac{5}{\sqrt{t'}}\sqrt{((\log\log 4t')_+ + \log(2Md/\delta))}\right) \leq \delta$$

*where $d_\infty(A, B) := \max_{i,j}|A_{i,j} - B_{i,j}|$.*

*Proof of Lemma 13.* We show the element-wise convergence of $\Sigma$ to $\hat{\Sigma}_t$ for the $(i, j)$ entry where $i, j = 1, \ldots, dM$. Consider the random sequence $X_s := \Sigma_{i,j} - \phi_i(\boldsymbol{x}_s)\phi_j(\boldsymbol{x}_s)$. We show that $X_1, \ldots, X_n$ satisfies conditions of Lemma 26. We first observe that

$$\mathbb{E}[X_s|X_{1:s-1}] = \mathbb{E}X_s = \Sigma(i, j) - \mathbb{E}_{\boldsymbol{x}\sim\pi}\phi_i(\boldsymbol{x})\phi_j(\boldsymbol{x}) = 0$$

since by definition $\Sigma_{i,j} = \mathbb{E}_{\boldsymbol{x}\sim\pi}\phi_i(\boldsymbol{x})^\top\phi_j(\boldsymbol{x})$. Moreover, we have normalized features $\|\boldsymbol{\phi}(\cdot)\| \leq 1$, therefore, each entry $\phi_i(\cdot)\phi_j(\cdot)$ is also bounded, yielding $|X_s| \leq 2$. Then Lemma 26 implies that for all $\tilde{\delta} > 0$,

$$\mathbb{P}\left(\exists t' : \; \frac{1}{t'}\sum_{s=1}^{t'} X_s \geq \frac{5}{\sqrt{t'}}\sqrt{\left((\log\log 4t')_+ + \log(2/\tilde{\delta})\right)}\right) \leq \tilde{\delta}.$$

Setting $\tilde{\delta} = \delta/(dM)$ and taking a union bound over all indices concludes the proof. $\square$

We are now ready to present the proofs to the lemmas in Appendix C.

*Proof of Lemma 7.* By (11) we have $\hat{\Sigma}_{t'} = \frac{\Phi_{\pi,t}^\top\Phi_{\pi,t}}{t'}$, and thereby

$$\kappa^2(\Phi_{\pi,t}, s) = \min_{b\in\Xi_s} b^\top\hat{\Sigma}_{t'}b = \tilde{\kappa}(\hat{\Sigma}_{t'}, s).$$

Inspired by Lemma 10.1 in van de Geer and Bühlmann [2009], we show that element-wise closeness of matrices $\Sigma$ and $\hat{\Sigma}_{t'}$ (c.f. Lemma 13) implies closeness in $\tilde{\kappa}$:

$$\left|\kappa^2(\Phi_{\pi,t}, s) - \tilde{\kappa}(\Sigma, s)\right| = \left|\tilde{\kappa}(\hat{\Sigma}_{t'}, s) - \tilde{\kappa}(\Sigma, s)\right|$$

$$= \left|\tilde{\kappa}\left(\hat{\Sigma}_{t'} - \Sigma, s\right)\right|$$

$$\leq \min_{b\in\Xi_s} d_\infty(\Sigma, \hat{\Sigma}_{t'})\|b\|_1^2$$

where the last line holds due to Hölder's. Moreover, since $b \in \Xi_s$, for any $J \subset [dM]$ where $|J| \leq s$ it additionally holds that $\|b_J\|_2 \leq 1$ and

$$\|b\|_1 \leq (1 + 3)\|b_J\|_1 \leq 4\sqrt{s}\|b_J\|_2 \leq 4\sqrt{s}$$

which gives,

$$\kappa^2(\Phi_{\pi,t}, s) \geq \tilde{\kappa}(\Sigma, s) - 16s\, d_\infty(\Sigma, \hat{\Sigma}_{t'}).$$

Therefore by Lemma 13,

$$\kappa^2(\Phi_{\pi,t}, s) \geq \tilde{\kappa}(\Sigma, s) - \frac{80s}{\sqrt{t'}}\sqrt{((\log\log 4t')_+ + \log(2Md/\delta))} \tag{12}$$

with probability greater than $1 - \delta$, simultaneously for all $t' \geq 1$. $\square$

*Proof of Lemma 8.* In Lemma 6 we showed that

$$\kappa^2(\Phi_t, s) \geq \frac{t'}{t}\kappa^2(\Phi_{\pi,t}, s)$$

where $t'$ indicates the number of rows in the exploratory sub-matrix of $\Phi_t$. Recall that $t' = \sum_{s=1}^t \alpha_s$ where $\alpha_s$ are i.i.d Bernoulli random variables with success probability of $\gamma_s$. Due to Lemma 24,

$$\mathbb{P}\left(\forall t \geq 1 : \; |t' - \Gamma_t| \leq \Delta_t\right) \geq 1 - \delta/2 \tag{13}$$

where

$$\Delta_t := \frac{5}{2}\sqrt{\frac{(\log\log t)_+ + \log(8/\delta)}{t}}, \quad \Gamma_t := \sum_{s=1}^{t} \gamma_s$$

Due to Lemma 7, with probability greater than $1 - \delta/2$ the following holds for all $t \geq 1$

$$\kappa^2(\Phi_t, 2) \geq \frac{t'}{t}\tilde{\kappa}(\Sigma, 2) - \frac{160\sqrt{t'}}{t}\sqrt{(\log\log 4t')_+ + \log(4Md/\delta)}$$

$$\geq \frac{\Gamma_t - \Delta_t}{t}\tilde{\kappa}(\Sigma, 2) - 160\sqrt{\frac{\Gamma_t + \Delta_t}{t^2}}\sqrt{(\log\log(4\Gamma_t + \Delta_t))_+ + \log(4Md/\delta)}$$

where the second inequality holds with probability $1 - \delta$, by incorporating (13) and taking a union bound. For the rest of the proof and to keep the calculations simple, we ignore the values of the absolute constants. We use the notation $g(t) = o(f(t))$ to show that $f(t)$ grows much faster than $g(t)$. More formally, if for every constant $c$ there exists $t_0$, where $g(t) \leq c|f(t)|$ for all $t \geq t_0$. Since $\gamma_s = \mathcal{O}(s^{-1/4})$ there exists $C$ such that $\Gamma_t = Ct^{3/4}$, then it is straightforward to observe that there exists absolute constants $\tilde{C}_i$ which satisfy,

$$\kappa^2(\Phi_t, 2) \geq \tilde{C}_1 t^{-1/4}\tilde{\kappa}(\Sigma, 2) - \frac{5t^{-3/2}\tilde{\kappa}(\Sigma, 2)}{2}\sqrt{(\log\log t)_+ + \log(8/\delta)}$$

$$- \tilde{C}_2 t^{-5/8}\sqrt{\log(Md/\delta) + (\log\log t)_+} - o\left(t^{-5/8}\sqrt{\log(Md/\delta) + (\log\log t)_+}\right)$$

$$\geq \tilde{C}_1 t^{-1/4}\tilde{\kappa}(\Sigma, 2) - \tilde{C}_3 t^{-5/8}\sqrt{\log(Md/\delta) + (\log\log t)_+}$$

The last inequality holds since $t^{-3/2}\sqrt{\log\log t} = o(t^{-5/8}\sqrt{\log\log t})$. The above chain of inequalities imply that there exist absolute constants $C_1, C_2$, for which

$$\mathbb{P}\left(\forall t \geq 1 : \kappa^2(\Phi_t, 2) \geq C_1\tilde{\kappa}(\Sigma, 2)t^{-1/4} - C_2 t^{-5/8}\sqrt{\log(Md/\delta) + (\log\log t)_+}\right) \geq 1 - \delta.$$

If $\pi$ is chosen according to (2), then $\tilde{\kappa}(\Sigma, 2) \geq C_{\min}$ yielding the lemma's second argument. $\qquad\square$

*Proof of Lemma 9.* Fix $j \in \{1, \ldots, M\}$, and construct the set

$$\Xi_{1,j} = \left\{\boldsymbol{b} \in \mathbb{R}^d \setminus \{0\} \Big| \boldsymbol{b} = (\boldsymbol{b}_1, \ldots, \boldsymbol{b}_M), \text{ s.t. } \boldsymbol{b}_j \in \mathbb{R}^d, \|\boldsymbol{b}_j\|_2 \leq 1 \text{ and } \forall j' \neq j : \boldsymbol{b}_{j'} = 0\right\}.$$

Note that $\Xi_{1,j} \subset \Xi_s$. Therefore,

$$\inf_{b \in \Xi_{1,j}} \|\Phi_t \boldsymbol{b}\|_2 \geq \inf_{b \in \Xi_s} \|\Phi_t \boldsymbol{b}\|_2 = \sqrt{t}\kappa(\Phi_t, s).$$

Moreover, by construction of $\Xi_{1,j}$ we have for all $\boldsymbol{b} \in \Xi_{1,j}$ that $\Phi_t \boldsymbol{b} = \Phi_{t,j}\boldsymbol{b}_j$, therefore,

$$\inf_{b \in \Xi_{1,j}} \|\Phi_t \boldsymbol{b}\|_2^2 = \inf_{\substack{\boldsymbol{b}_j \in \mathbb{R}^d \\ \|\boldsymbol{b}_j\|_2^2 \leq 1}} \|\Phi_{t,j}\boldsymbol{b}_j\|_2^2 = \lambda_{\min}(\Phi_{t,j}^\top \Phi_{t,j}).$$

From the above equations we conclude that $\lambda_{\min}(\Phi_{t,j}^\top \Phi_{t,j}) \geq t\kappa^2(\Phi_t, s)$, for all $j = 1, \ldots, M$. Therefore, using Lemma 8 we obtain that there exists $C_1, C_2$ such that

$$\mathbb{P}\left(\forall t \geq 1, j = 1, \ldots, M : \lambda_{\min}(\Phi_{t,j}^\top \Phi_{t,j}) \geq C_1 C_{\min} t^{3/4} - C_2 t^{3/8}\sqrt{\log(Md/\delta) + (\log\log t)_+}\right) \geq 1 - \delta$$

$\qquad\square$

*Proof of Proposition 10.* Since $\mathcal{X}$ is a convex body, then there exists an invertible map $T$, such that $T(\mathcal{X})$ is an isotropic body [e.g. Proposition 1.1.1., Giannopoulos, 2003]. Then by definition, $\bar{X} \sim \text{Unif}(T(\mathcal{X}))$ is an isotropic distribution and $\text{Cov}(\bar{X}) = I_d$ [e.g., c.f. Chapter 3.3.5 Vershynin, 2018]. Since $\phi$ is linear and invertible, it may be written is as $\phi(\boldsymbol{x}) = A\boldsymbol{x}$, where $A$ is an invertible matrix. Therefore,

$$\Sigma(\pi, \boldsymbol{\phi}) = \text{Cov}(\boldsymbol{\phi}(X)) = A^\top\text{Cov}(X)A = A^\top\text{Cov}\left(T^{-1}\bar{X}\right)A = A^\top(T^{-1})^2 A.$$

As for the minimum eigenvalue, suppose $\boldsymbol{v} \in \mathbb{R}^d$ and $\|\boldsymbol{v}\| = 1$, then

$$C_{\min} \geq \lambda_{\min}(\Sigma(\pi, \boldsymbol{\phi})) \geq \boldsymbol{v}^\top A^\top(T^{-1})^2 A\boldsymbol{v} \geq \|A\boldsymbol{v}\|_2 \lambda_{\min}(T^{-2}) = \frac{\|A\boldsymbol{v}\|_2}{\lambda_{\max}^2(T)} \geq \frac{\lambda_{\min}(A)}{\lambda_{\max}^2(T)}.$$

$\qquad\square$

*Proof of Proposition 11.* By the assumption of the proposition, for all $i \in M$

$$[\Sigma(\pi, \boldsymbol{\phi})]_{i,i} = \mathbb{E}_{\boldsymbol{x} \sim \pi} \phi_i^2(\boldsymbol{x}) = \frac{1}{\text{Vol}(\mathcal{X})} \int_{\mathcal{X}} \phi_i^2(\boldsymbol{x}) \mathrm{d}\mu(\boldsymbol{x}) \geq 1$$

and for all $i \neq j$,

$$[\Sigma(\pi, \boldsymbol{\phi})]_{i,j} = \mathbb{E}_{\boldsymbol{x} \sim \pi} \phi_i(\boldsymbol{x})\phi_j(\boldsymbol{x}) = \frac{1}{\text{Vol}(\mathcal{X})} \int_{\mathcal{X}} \phi_i(\boldsymbol{x})\phi_j(\boldsymbol{x}) \mathrm{d}\mu(\boldsymbol{x}) = 0$$

We use $\Sigma = \Sigma(\pi, \boldsymbol{\phi})$. For any $\boldsymbol{b} \in \mathbb{R}^{Md}$ where $\|\boldsymbol{b}\| \leq 1$,

$$\boldsymbol{b}^\top \Sigma \boldsymbol{b} = \sum_{i,j \in [M]} \boldsymbol{b}_j^\top \Sigma_{i,j} \boldsymbol{b}_i = \sum_{i \in [M]} \boldsymbol{b}_i^\top \Sigma_{i,i} \boldsymbol{b}_i + \sum_{i,j \in [M], i \neq j} \boldsymbol{b}_j^\top \Sigma_{i,j} \boldsymbol{b}_i$$

$$= \sum_{i \in [M]} \boldsymbol{b}_i^\top \Sigma_{i,i} \boldsymbol{b}_i$$

$$\geq 1 \sum_{i \in [M]} \|\boldsymbol{b}_i\|_2^2 \geq 1.$$

Which implies,

$$\tilde{\kappa}(\Sigma, s) = \min_{\boldsymbol{b} \in \Xi_s} \boldsymbol{b}^\top \Sigma \boldsymbol{b} \geq 1.$$

By Lemma 8, there exist absolute constants $C_1$ and $C_2$ for which,

$$\mathbb{P}\left(\forall t \geq 1 : \ \kappa^2(\Phi_t, 2) \geq \tilde{\kappa}(\Sigma, 2)C_1 t^{-1/4} - C_2 t^{-5/8} \sqrt{\log(Md/\delta) + (\log\log t)_+}\right) \geq 1 - \delta.$$

concluding the proof.

$\square$

# D  Proof of Regret Bound

**Theorem 14** (Anytime Regret, Formal). *Let $\delta \in (0, 1]$ and $\pi$ be the maximizer of (2). Assume the oracle agent employs a UCB or a Greedy policy, as laid out in Section 5. Suppose $\eta_t = \mathcal{O}(C_{\min} t^{-1/2} C(M, \delta, d))$ and $\gamma_t = \mathcal{O}(t^{-1/4})$ and $\lambda_t = \mathcal{O}(C(M, \delta, d)t^{-1/2})$, then exists absolute constants $C_1, \ldots, C_6$ for which* ALEXP *attains the regret*

$$R(n) \leq C_1 B n^{3/4} + C_2 \sqrt{n} C_{\min}^{-1} C(M, \delta, d) \log M$$
$$+ C_3 B^2 C_{\min} \sqrt{n} + C_4 B \sqrt{n \left((\log\log nB^2)_+ + \log(1/\delta)\right)}$$
$$+ C_5 \left(1 + C_{\min}^{-1} n^{-3/8} \sqrt{\log(Md/\delta) + (\log\log n)_+}\right)$$
$$\times \left[B n^{1/4} + (n^{3/4} + \frac{\log n}{C_{\min}})C(M, \delta, d) + \frac{n^{5/8}}{\sqrt{C_{\min}}} \sqrt{d \log n + \log(1/\delta) + B^2}\right]$$

*with probability greater than $1 - \delta$, simultaneously for all $n \geq 1$. Here,*

$$C(M, \delta, d) = C_6 \sigma \sqrt{1 + \sqrt{d \left(\log(M/\delta) + (\log\log d)_+\right)} + (\log(M/\delta) + (\log\log d)_+)}.$$

Our main regret bound is an immediate corollary of Lemma 15 and Lemma 16, considering the regret decomposition of (3).

**Lemma 15** (Virtual Regret of the Oracle). *Let $\delta \in (0, 1]$ and $\tilde{\lambda} > 0$. Assume the oracle agent employs a UCB or a Greedy policy, as laid out in Section 5. If $\gamma_t = \mathcal{O}(t^{1/4})$, there exists an absolute constant $C_1$ for which with probability greater than $1 - \delta$, simultaneously for all $n \geq 1$,*

$$\tilde{R}_{j^\star}(n) = \frac{C_1 n^{5/8}}{\sqrt{C_{\min}}} \left(1 + n^{-3/8} C_{\min}^{-1} \sqrt{\log(Md/\delta) + (\log\log n)_+}\right)$$

$$\times \sqrt{\sigma^2 d \log\left(\frac{n}{\tilde{\lambda}d} + 1\right) + 2\sigma^2 \log(1/\delta) + \tilde{\lambda}B^2}$$

**Lemma 16** (Any-Time Model-Selection Regret, Formal). *Let $\delta \in (0,1]$ and $\pi$ be the maximizer of (2). Suppose $\eta_t = \mathcal{O}(C_{\min}/\sqrt{t}C(M,\delta,d))$ and $\gamma_t = \mathcal{O}(t^{-1/4})$ and $\lambda_t = \mathcal{O}(C(M,\delta,d)/\sqrt{t})$, then exists absolute constants $C_i$ for which* ALEXP *attains the model selection regret*

$$R(n,j) \le C_1 B n^{3/4} + C_2 \sqrt{n} C_{\min}^{-1} C(M,\delta,d) \log M$$
$$+ C_3 B^2 C_{\min} \sqrt{n} + C_4 B \sqrt{n \left((\log \log n B^2)_+ + \log(1/\delta)\right)}$$
$$+ C_5 \left( B n^{1/4} + (n^{3/4} + \frac{\log n}{C_{\min}}) C(M,\delta,d) \right) \left( 1 + C_{\min}^{-1} n^{-3/8} \sqrt{\log(Md/\delta) + (\log \log n)_+} \right)$$

*with probability greater than $1 - \delta$, simultaneously for all $n \ge 1$. Here,*

$$C(M,\delta,d) = C_6 \sigma \sqrt{1 + \sqrt{d \left(\log(M/\delta) + (\log \log d)_+\right)} + (\log(M/\delta) + (\log \log d)_+)}.$$

### D.1 Proof of Model Selection Regret

Our technique for bounding the model selection regret relies on a classic horizon-independent analysis of the exponential weights algorithm, presented in Lemma 17.

**Lemma 17** (Anytime Exponential Weights Guarantee). *Assume $\eta_t \hat{r}_{t,j} \le 1$ for all $1 \le j \le M$ and $t \ge 1$. If the sequence $(\eta_t)_{t \ge 1}$ is non-increasing, then for all $n \ge 1$,*

$$\sum_{t=1}^{n} \hat{r}_{t,k} - \sum_{t=1}^{n} \sum_{j=1}^{M} q_{t,j} \hat{r}_{t,j} \le \frac{\log M}{\eta_n} + \sum_{t=1}^{n} \eta_t \sum_{j=1}^{M} q_{t,j} \hat{r}_{t,j}^2$$

*for any arm $k \in [M]$.*

*Proof of Lemma 17.* Define $\hat{R}_{t,i} := \sum_{s=1}^{t} \hat{r}_{s,i}$ to be the expected cumulative reward of agent $i$ after $t$ steps. We rewrite for a fixed $k$

$$\sum_{t=1}^{n} \hat{r}_{t,k} - \sum_{t=1}^{n} \sum_{j=1}^{M} q_{t,j} \hat{r}_{t,j} = \sum_{t=1}^{n} \hat{r}_{t,k} - \sum_{t=1}^{n} \mathbb{E}_{j \sim q_t}[\hat{r}_{t,j}]. \tag{14}$$

We focus on a single term in the second sum. For any $t$, we have

$$-\mathbb{E}_{j \sim q_t}[\hat{r}_{t,j}] = \log(\exp(-\mathbb{E}_{j \sim q_t}[\frac{\eta_t}{\eta_t} \hat{r}_{t,j}])) = \log(\exp(-\mathbb{E}_{j \sim q_t}[\eta_t \hat{r}_{t,j}])^{1/\eta_t})$$

$$= \frac{1}{\eta_t} \log(\exp(-\mathbb{E}_{j \sim q_t}[\eta_t \hat{r}_{t,j}]))$$

$$= \frac{1}{\eta_t} \log(\mathbb{E}_{i \sim q_t} \exp(-\mathbb{E}_{j \sim q_t}[\eta_t \hat{r}_{t,j}])) \tag{15}$$

The inner expectation is over $j$, while the outer one is over $i$ and therefore has no effect. Moreover,

$$\frac{1}{\eta_t} \log \mathbb{E}_{i \sim q_t} \exp(-\eta_t \mathbb{E}_{j \sim q_t}[\hat{r}_{t,j}] + \eta_t \hat{r}_{t,i}) = \frac{1}{\eta_t} \log \left(\exp(-\eta_t \mathbb{E}_{j \sim q_t}[\hat{r}_{t,j}]) \mathbb{E}_{i \sim q_t} \exp(\eta_t \hat{r}_{t,i})\right)$$

$$= \frac{1}{\eta_t} \log \mathbb{E}_{i \sim q_t} \exp(-\eta_t \mathbb{E}_{j \sim q_t}[\hat{r}_{t,j}])$$

$$+ \frac{1}{\eta_t} \log \mathbb{E}_{i \sim q_t} \exp(\eta_t \hat{r}_{t,i}) \tag{16}$$

where again, the expectation can be reintroduced to get the last line. Combining (15) and (16),

$$-\mathbb{E}_{j \sim q_t}[\hat{r}_{t,j}] = \frac{1}{\eta_t} \log \mathbb{E}_{i \sim q_t} \exp(-\eta_t \mathbb{E}_{j \sim q_t}[\hat{r}_{t,j}] + \eta_t \hat{r}_{t,i}) - \frac{1}{\eta_t} \log \mathbb{E}_{i \sim q_t} \exp(\eta_t \hat{r}_{t,i}) \tag{17}$$

This transformation is at the core of many exponential weight proofs [Bubeck et al., 2012, Lattimore and Szepesvári, 2020]. We first bound the first term in (17):

$$\log \mathbb{E}_{i \sim q_t} \exp(-\eta_t \mathbb{E}_{j \sim q_t}[\hat{r}_{t,j}] + \eta_t \hat{r}_{t,i}) = \log \mathbb{E}_{i \sim q_t} \exp(\eta_t \hat{r}_{t,i}) - \eta_t \mathbb{E}_{j \sim q_t} \hat{r}_{t,j}$$

$$\overset{(I)}{\leq} \mathbb{E}_{i \sim q_t} \exp(\eta_t \hat{r}_{t,i}) - 1 - \eta_t \mathbb{E}_{j \sim q_t} \hat{r}_{t,j}$$

$$= \mathbb{E}_{i \sim q_t} \left[ \exp(\eta_t \hat{r}_{t,i}) - 1 - \eta_t \hat{r}_{t,i} \right]$$

$$\overset{(II)}{\leq} \mathbb{E}_{i \sim q_t} \left[ \eta_t^2 \hat{r}_{t,i}^2 \right] \tag{18}$$

where in (I) we use the fact that $\log(z) \leq z - 1$ and in (II) we use the fact that for $x \leq 1$, we have $\exp(x) \leq 1 + x + x^2$, and hence $\exp(x) - 1 - x \leq x^2$. For the second term in (17), we will mirror the potential argument in Bubeck et al. [2012], but with a slightly different potential function. We expand the definition of $q_t$:

$$-\frac{1}{\eta_t} \log \mathbb{E}_{i \sim q_t} \exp(\eta_t \hat{r}_{t,i}) = -\frac{1}{\eta_t} \log \frac{\sum_{i=1}^{M} \exp(\eta_t \hat{R}_{t,i})}{\sum_{i=1}^{M} \exp(\eta_t \hat{R}_{t-1,i})}$$

$$= -\frac{1}{\eta_t} \log \frac{1}{M} \sum_{i=1}^{M} \exp(\eta_t \hat{R}_{t,i}) + \frac{1}{\eta_t} \log \frac{1}{M} \sum_{i=1}^{M} \exp(\eta_t \hat{R}_{t-1,i})$$

$$= J_t(\eta_t) - J_{t-1}(\eta_t), \tag{19}$$

where we define $J_t(\eta) = -\frac{1}{\eta} \log \frac{1}{M} \sum_{i=1}^{M} \exp(\eta \hat{R}_{t,i})$. We also define $F_t(\eta) = \frac{1}{\eta} \log \frac{1}{M} \sum_{i=1}^{M} \exp(-\eta \hat{R}_{t,i})$. We observe the relation $J(\eta) = F(-\eta)$. From this, it follows that for any $\eta$, we have $J'(\eta) = -F'(-\eta) \leq 0$, by the argument in Bubeck et al. [2012, Theorem 3.1] that shows $F'(\eta) \geq 0$ for any $\eta$.

**Putting together the pieces** Now, we can bound (17) by inputing (18) and (19):

$$-\mathbb{E}_{j \sim q_t}[\hat{r}_{t,j}] \leq \mathbb{E}_{i \sim q_t} \left[ \eta_t \hat{r}_{t,i}^2 \right] + J_t(\eta_t) - J_{t-1}(\eta_t)$$

With this, we rewrite (14) as

$$\sum_{t=1}^{n} \hat{r}_{t,k} - \sum_{t=1}^{n} \mathbb{E}_{j \sim q_t}[\hat{r}_{t,j}] = \sum_{t=1}^{n} \hat{r}_{t,k} + \sum_{t=1}^{n} \mathbb{E}_{i \sim q_t} \left[ \eta_t \hat{r}_{t,i}^2 \right] + \sum_{t=1}^{n} J_t(\eta_t) - J_{t-1}(\eta_t) \tag{20}$$

**Potential manipulation** We can do an Abel transformation on the sum of potentials in (20), namely obtaining

$$\sum_{t=1}^{n} J_t(\eta_t) - J_{t-1}(\eta_t) = \sum_{t=1}^{n-1} (J_t(\eta_t) - J_t(\eta_{t+1})) + J_n(\eta_n),$$

where we used that $J_0(\eta) = 0$. We know $J'(\eta) \leq 0$ and so $J$ is decreasing and since $\eta_{t+1} \leq \eta_t$, we have $J(\eta_{t+1}) \geq J(\eta_t)$ or $(J_t(\eta_t) - J_t(\eta_{t+1})) \leq 0$, so that for any fixed $k$

$$\sum_{t=1}^{n} J_t(\eta_t) - J_{t-1}(\eta_t) \leq J_n(\eta_n) \leq \frac{\log(M)}{\eta_n} - \frac{1}{\eta_n} \log \left( \sum_{i=1}^{M} \exp(\eta_n \hat{R}_{n,i}) \right)$$

$$\overset{(*)}{\leq} \frac{\log(M)}{\eta_n} - \frac{1}{\eta_n} \log \left( \exp(\eta_n \hat{R}_{n,k}) \right)$$

$$= \frac{\log(M)}{\eta_n} - \sum_{t=1}^{n} \hat{r}_{t,k} \tag{21}$$

where $(*)$ follows because $\exp$ is positive and $-\log$ is decreasing (notice that we drop $M - 1$ terms from the sum). Plugging (21) into (20), we obtain

$$\sum_{t=1}^{n} \hat{r}_{t,k} - \sum_{t=1}^{n} \mathbb{E}_{j \sim q_t}[\hat{r}_{t,j}] \leq \sum_{t=1}^{n} \hat{r}_{t,k} + \sum_{t=1}^{n} \mathbb{E}_{i \sim q_t} \left[ \eta_t \hat{r}_{t,i}^2 \right] + \sum_{t=1}^{n} J_t(\eta_t) - J_{t-1}(\eta_t)$$

$$\leq \sum_{t=1}^{n} \hat{r}_{t,k} + \sum_{t=1}^{n} \mathbb{E}_{i \sim q_t} \left[ \eta_t \hat{r}_{t,i}^2 \right] + \frac{\log(M)}{\eta_n} - \sum_{t=1}^{n} \hat{r}_{t,k}$$

$$\leq \sum_{t=1}^{n} \mathbb{E}_{i \sim q_t} \left[ \eta_t \hat{r}_{t,i}^2 \right] + \frac{\log(M)}{\eta_n}$$

$$= \sum_{t=1}^{n} \eta_t \sum_{j=1}^{M} q_{t,j} \hat{r}_{t,j}^2 + \frac{\log(M)}{\eta_n}.$$

$\square$

We expressed in Section 5.2, that the model selection regret of ALEXP, is closely tied to the bias and variance of the reward estimates $\hat{r}_{t,j}$. The following lemma formalizes this claim.

**Lemma 18.** *(Anytime Generic regret bound) If $\eta_t$ is picked such that $\eta_t \hat{r}_{t,j} \leq 1$ for all $1 \leq j \leq M$ and $1 \leq t$ almost surely, then Algorithm 1 satisfies with probability greater than $1 - 2\delta/3$, that simultaneously for all $n \geq 1$*

$$R(n,i) \leq 2B \sum_{t=1}^{n} \gamma_t + \frac{\log M}{\eta_n} + \sum_{t=1}^{n} \eta_t \sum_{j=1}^{M} q_{t,j} \hat{r}_{t,j}^2 + \sum_{t=1}^{n} (\omega_{t,i} + \sum_{j=1}^{M} q_{t,j} \omega_{t,j})$$
$$+ 10B \sqrt{n \left( (\log \log nB^2)_+ + \log(12/\delta) \right)}$$

*where $\omega_{t,i} = |r_{t,i} - \hat{r}_{t,i}|$.*

*Proof of Lemma 18.* Let $\alpha_t$ denote the Bernoulli random variable that is equal to 1 if at step $t$ we select actions according to $\pi$ and 0 otherwise. At each step $t$ with $\alpha_t = 1$ ALEXP accumulates a regret of at most $2B$, since $\|\boldsymbol{\theta}\|_\infty \leq B$ and $\|\boldsymbol{\phi}(\cdot)\| \leq 1$. We can decompose the regret as,

$$R(n,i) \leq \sum_{t=1}^{n} 2B\alpha_t + (r_{t,i} - r_t)(1 - \alpha_t)$$

For the first term, by Lemma 24, we have

$$2B \sum_{t=1}^{n} \alpha_t \leq 2B \left( \sum_{t=1}^{n} \gamma_t + \frac{5}{2} \sqrt{n \left( (\log \log n)_+ + \log(4/\delta_1) \right)} \right).$$

simultaneously for all $n \geq 1$, with probability $1 - \delta_1$. Let $\hat{r}_t := \sum_{j=1}^{M} q_{t,j} \hat{r}_{t,j}$. We may re-write the second term of the regret as follows,

$$\sum_{t=1}^{n} (1 - \alpha_t) \left( r_{t,i} - r_t \right) \leq \sum_{t=1}^{n} (1 - \alpha_t) \left[ (r_{t,i} - \hat{r}_{t,i}) + (\hat{r}_{t,i} - \hat{r}_t) + (\hat{r}_t - r_t) \right]$$
$$\leq \sum_{t=1}^{n} \omega_{t,i} + (1 - \alpha_t) \left[ (\hat{r}_{t,i} - \hat{r}_t) + (\hat{r}_t - r_t) \right]$$

We bound the second term on the right hand side, using Lemma 17

$$\sum_{t=1}^{n} (1 - \alpha_t)(\hat{r}_{t,i} - \hat{r}_t) \leq \sum_{t=1}^{n} (\hat{r}_{t,i} - \hat{r}_t) \leq \frac{\log M}{\eta_n} + \sum_{t=1}^{n} \eta_t \sum_{j=1}^{M} q_{t,j} \hat{r}_{t,j}^2.$$

As for the third term,

$$(1 - \alpha_t)(\sum_{j=1}^{M} q_{t,j} \hat{r}_{t,j} - r_t) = (1 - \alpha_t) \left[ \sum_{j=1}^{M} q_{t,j} (\hat{r}_{t,j} - r_{t,j} + r_{t,j}) - r_t \right]$$
$$\leq \sum_{j=1}^{M} q_{t,j} \omega_{t,j} + (1 - \alpha_t) \left( r_t - \sum_{j=1}^{M} q_{t,j} r_{t,j} \right).$$

It remains to bound the deviation term. For all $t$ that satisfy $\alpha_t = 0$, the action/model is selected according to $q_{t,j}$, therefore the conditional expectation of $r_t$ can be written as

$$\mathbb{E}_{t-1} r_t = \sum_{j-1}^{M} q_{t,j} r_{t,j}$$

The sequence $X_t := r_t - \mathbb{E}_{t-1} r_t$ is a martingale difference sequence adapted to the history $H_t$, since for every $t \geq 1$,

$$\mathbb{E}_{t-1} X_t = \mathbb{E}\left[r_t - \mathbb{E}_{t-1} r_t | H_{t-1}\right] = 0.$$

Since $r_t \leq B$, then $X_t \leq 2B$ almost surely, which allows for an application of anytime Azuma-Hoeffding (Lemma 26):

$$\mathbb{P}\left(\exists n: \sum_{t=1}^{n}\left(r_t - \sum_{j=1}^{M} q_{t,j} r_{t,j}\right) \geq \frac{5B}{2}\sqrt{n\left(\left(\log\log nB^2\right)_+ + \log(2/\delta_2)\right)}\right) \leq \delta_2$$

which, in turn, leads us to

$$\sum_{t=1}^{n}(1-\alpha_t)\left(r_t - \sum_{j=1}^{M} q_{t,j} r_{t,j}\right) \overset{\text{a.s.}}{\leq} \sum_{t=1}^{n}\left(r_t - \sum_{j=1}^{M} q_{t,j} r_{t,j}\right)$$

$$\overset{\text{w.h.p.}}{\leq} \frac{5B}{2}\sqrt{n\left(\left(\log\log nB^2\right)_+ + \log(2/\delta_2)\right)}$$

simultaneously for all $n \geq 1$. We set $\delta_1 = \delta_2 = \delta/3$, take a union bound and put the terms together obtaining,

$$R(n,i) \leq 2B\sum_{t=1}^{n}\gamma_t + \frac{\log M}{\eta n} + \eta\sum_{t=1}^{n}\sum_{j=1}^{M} q_{t,j}\hat{r}_{t,j}^2 + \sum_{t=1}^{n}\left(\omega_{t,i} + \sum_{j=1}^{M} q_{t,j}\omega_{t,j}\right)$$

$$+ \frac{5B}{2}\sqrt{n\left(\left(\log\log nB^2\right)_+ + \log(6/\delta)\right)} + 5B\sqrt{n\left(\left(\log\log n\right)_+ + \log(12/\delta)\right)}$$

We upper bound the sum of last two terms to conclude the proof. $\qquad\square$

The next two lemmas bound the bias and variance terms which appear in Lemma 18.

**Lemma 19** (Anytime Bound on the Bias Term). *If the regularization parameter of Lasso is chosen at every step as*

$$\lambda_t = \frac{2\sigma}{\sqrt{t}}\sqrt{1 + \frac{5}{\sqrt{2}}\sqrt{d\left(\log(2M/\delta) + (\log\log d)_+\right)} + \frac{12}{\sqrt{2}}\left(\log(2M/\delta) + (\log\log d)_+\right)}$$

*and $\gamma_t = O(t^{-1/4})$, then with probability greater than $1-\delta$, simultaneously for all $n \geq 1$,*

$$\sum_{t=1}^{n}|\hat{r}_{t,i} - r_{t,i}| \leq n^{3/4}C_{\min}^{-1}C(M,\delta,d)\left(1 + n^{-3/8}C_{\min}^{-1}\sqrt{\log(Md/\delta) + (\log\log n)_+}\right)$$

*where*

$$C(M,\delta,d) := C\sigma\sqrt{1 + \sqrt{d\left(\log(M/\delta) + (\log\log d)_+\right)} + (\log(M/\delta) + (\log\log d)_+)}$$

*and $C$ is an absolute constant.*

*Proof of Lemma 19.* By the definition of the expected reward and its estimate,

$$\sum_{t=1}^{n}|\hat{r}_{t,i} - r_{t,i}| = \sum_{t=1}^{n}\left|\int_{\mathcal{X}}(r(\boldsymbol{x}) - \hat{r}_t(\boldsymbol{x}))\mathrm{d}p_{t+1,i}(\boldsymbol{x})\right|$$

$$\leq \sum_{t=1}^{n}\int_{\mathcal{X}}|r(\boldsymbol{x}) - \hat{r}_t(\boldsymbol{x})|\mathrm{d}p_{t+1,i}(\boldsymbol{x})$$

$$\overset{\text{C.S.}}{\leq} \sum_{t=1}^{n}\int_{\mathcal{X}}\left\|\boldsymbol{\theta} - \hat{\boldsymbol{\theta}}_t\right\|_2\|\boldsymbol{\phi}(\boldsymbol{x})\|_2\mathrm{d}p_{t+1,i}(\boldsymbol{x})$$

$$\overset{\text{bdd. }\phi}{\leq} \sum_{t=1}^{n}\left\|\boldsymbol{\theta} - \hat{\boldsymbol{\theta}}_t\right\|_2\int_{\mathcal{X}}\mathrm{d}p_{t+1,i}(\boldsymbol{x}) = \sum_{t=1}^{n}\left\|\boldsymbol{\theta} - \hat{\boldsymbol{\theta}}_t\right\|_2.$$

We highlight that the Cauchy-Schwarz step may be refined. By further assuming that $\boldsymbol{\theta}_{j^\star}$ is bounded away from zero (also called the *beta-min* condition [Bühlmann and Van De Geer, 2011]) one can show that $\boldsymbol{\theta} - \hat{\boldsymbol{\theta}}_t$ is a 2-sparse vector. This will then allow one to only rely on boundedness of $\|\boldsymbol{\phi}_j\|$ rather than $\|\boldsymbol{\phi}\|$ to derive the last inequality, and relax our assumption of $\|\boldsymbol{\phi}(\cdot)\| \leq 1$ to $\|\boldsymbol{\phi}_j(\cdot)\| \leq 1$ for all $j \in [M]$. From Theorem 3, with probability greater than $1 - \delta/2$ simultaneously for all $n \geq 1$,

$$\sum_{t=1}^n |\hat{r}_{t,i} - r_{t,i}| \leq \sum_{t=1}^n \frac{4\sqrt{10}\lambda_t}{\kappa^2(\Phi_t, 2)} = \tilde{C}(M, \delta, d) \sum_{t=1}^n \frac{1}{\kappa^2(\Phi_t, 2)\sqrt{t}}$$

where,

$$\tilde{C}(M, \delta, d) := 8\sigma\sqrt{1 + \frac{5}{\sqrt{2}}\sqrt{d\left(\log(4M/\delta) + (\log\log d)_+\right)} + \frac{12}{\sqrt{2}}\left(\log(4M/\delta) + (\log\log d)_+\right)}.$$

From Lemma 8, there exist absolute constants $C_1, C_2$ for which,

$$\mathbb{P}\left(\forall t \geq 1: \ \kappa^2(\Phi_t, 2) \geq C_1 C_{\min} t^{-1/4} - C_2 t^{-5/8}\sqrt{\log(Md/\delta) + (\log\log t)_+}\right) \geq 1 - \delta.$$

Using Taylor approximation we observe that, $\frac{1}{1-x^{-1}} = 1 + x^{-1} + o(x^{-1}) = \mathcal{O}(1 + x^{-1})$. Therefore, these exists absolute constant $C_3, C_4$, for which with probability greater than $1 - \delta$ for all $t \geq 1$

$$\sum_{t=1}^n \frac{\tilde{C}(M, \delta, d)}{\kappa^2(\Phi_t, 2)\sqrt{t}} \leq \sum_{t=1}^n \frac{\tilde{C}(M, \delta, d)}{\sqrt{t}} \frac{1}{C_1 C_{\min} t^{-1/4} - C_2 t^{-5/8}\sqrt{\log(Md/\delta) + (\log\log t)_+}}$$

$$\leq \sum_{t=1}^n \frac{\tilde{C}(M, \delta, d)}{\sqrt{t}} \frac{\tilde{C}_3}{C_{\min} t^{-1/4}}\left(1 + \frac{C_2 t^{-5/8}\sqrt{\log(Md/\delta) + (\log\log t)_+}}{C_1 C_{\min} t^{-1/4}}\right)$$

$$\leq \sum_{t=1}^n \frac{\tilde{C}_3 \tilde{C}(M, \delta, d) t^{-1/4}}{C_{\min}}\left(1 + \tilde{C}_4 \frac{t^{-3/8}}{C_{\min}}\sqrt{\log(Md/\delta) + (\log\log t)_+}\right)$$

$$= \frac{C_3 \tilde{C}(M, \delta, d) n^{3/4}}{C_{\min}}\left(1 + C_4 \frac{n^{-3/8}}{C_{\min}}\sqrt{\log(Md/\delta) + (\log\log n)_+}\right).$$

$\square$

**Lemma 20** (Anytime Bound on Variance Term). *Suppose $\lambda_t$ is chosen according to Lemma 19, $\gamma_t = \mathcal{O}(t^{-1/4})$ and $\eta_t = \mathcal{O}(C_{\min} t^{-1/2}/C(M, \delta, d))$. Then with probability greater than $1 - \delta$, the following holds simultaneously for all $n \geq 1$ and $t \geq 1$*

$$\hat{r}_{t,j} \leq \frac{4\sqrt{10}\lambda_t}{\kappa^2(\Phi_t, 2)} + B, \qquad \forall j \in [M]$$

$$\sum_{t=1}^n \eta_t \sum_{j=1}^M q_{t,j}\hat{r}_{t,j}^2 \leq C_1 B^2 C_{\min}\sqrt{n} + C_2 B n^{1/4}\left(1 + \frac{n^{-3/8}}{C_{\min}}\sqrt{\log(Md/\delta) + (\log\log n)_+}\right)$$

$$+ C(M, \delta, d)\frac{\log n}{C_{\min}}\left(1 + \frac{n^{-3/8}}{C_{\min}}\sqrt{\log(Md/\delta) + (\log\log n)_+}\right)$$

*where $C_i$ are absolute constants, and $C(M, \delta, d)$ is as defined in Lemma 19, up to constant factors.*

*Proof of Lemma 20.* We start by upper bounding $\hat{r}_{t,j}$. For all $j$ and $t$ it holds that:

$$\hat{r}_{t,j} = \int_{\mathcal{X}} \langle \hat{\boldsymbol{\theta}}_t, \boldsymbol{\phi}(\boldsymbol{x})\rangle \mathrm{d}p_{t+1,j}(\boldsymbol{x}) \leq \left\|\hat{\boldsymbol{\theta}}_t\right\| \int_{\mathcal{X}} \|\boldsymbol{\phi}(\boldsymbol{x})\|\mathrm{d}p_{t+1,j}(\boldsymbol{x}) \leq \left\|\hat{\boldsymbol{\theta}}_t\right\|_2 \leq B + \left\|(\hat{\boldsymbol{\theta}}_t - \boldsymbol{\theta})\right\|_2$$

since $\|\boldsymbol{\theta}\|_2 \leq B$. To bound the last term, we only need to invoke Theorem 3, which, in turn, will simultaneously bound $\hat{r}_{t,j}$ for all $j = 1, \ldots, M$:

$$\mathbb{P}\left(\forall t \geq 1, \forall j \in [M]: \ \hat{r}_{t,j} \leq \frac{4\sqrt{10}\lambda_t}{c_{\kappa,t}^2} + B\right) \geq 1 - \delta$$

Which implies for all $t \geq 1$,

$$\sum_{j=1}^{M} q_{t,j} \hat{r}_{t,j}^2 \leq \left( \frac{4\sqrt{10}\lambda_t}{c_{\kappa,t}^2} + B \right)^2 \sum_{j=1}^{M} q_{t,j} = \frac{160\lambda_t^2}{c_{\kappa,t}^4} + B^2 + \frac{8B\sqrt{10}\lambda_t}{c_{\kappa,t}^2}.$$

For the last term, similar to the proof of Lemma 19 we have,

$$\sum_{t=1}^{n} \eta_t \frac{8B\sqrt{10}\lambda_t}{c_{\kappa,t}^2} \leq C_1 B n^{1/4} \left( 1 + \frac{n^{-3/8}}{C_{\min}} \sqrt{\log(Md/\delta) + (\log\log n)_+} \right)$$

for some absolute constant $C_1$. We treat the squared term similarly,

$$\sum_{t=1}^{n} \eta_t \frac{160\lambda_t^2}{c_{\kappa,t}^4} \leq C(M,\delta,d) \sum_{t=1}^{n} \frac{\bar{C}_3}{tC_{\min}} \left( 1 + \bar{C}_4 \frac{t^{-3/8}}{C_{\min}} \sqrt{\log(Md/\delta) + (\log\log t)_+} \right)$$

$$\leq C(M,\delta,d) \frac{C_3 \log n}{C_{\min}} \left( 1 + C_4 \frac{n^{-3/8}}{C_{\min}} \sqrt{\log(Md/\delta) + (\log\log n)_+} \right).$$

Note that the last inequality is not tight. This term will not be fastest growing term in the regret, so we have little motivation to bound it tightly. Therefore,

$$\sum_{t=1}^{n} \eta_t \sum_{j=1}^{M} q_{t,j} \hat{r}_{t,j}^2 \leq C_1 B^2 C_{\min} \sqrt{n} + C_2 B n^{1/4} \left( 1 + \frac{n^{-3/8}}{C_{\min}} \sqrt{\log(Md/\delta) + (\log\log t)_+} \right)$$

$$+ C(M,\delta,d) \frac{\log n}{C_{\min}} \left( 1 + C_4 \frac{n^{-3/8}}{C_{\min}} \sqrt{\log(Md/\delta) + (\log\log n)_+} \right)$$

where $C_i$ are absolute constants. $\qquad\square$

***Proof of Lemma 16.*** We start by conditioning on the event $E$ that $\eta_t$ is picked such that $\eta_t \hat{r}_{t,j} \leq 1$ for all $t \geq 1$ and $j = 1, \ldots, M$. Then by application of Lemma 18 we get with probability greater than $1 - 2\delta/3$,

$$R(n,i) \leq 2B \sum_{t=1}^{n} \gamma_t + \frac{\log M}{\eta_n} + \sum_{t=1}^{n} \eta_t \sum_{j=1}^{M} q_{t,j} \hat{r}_{t,j}^2 + \sum_{t=1}^{n} \left( \omega_{t,i} + \sum_{j=1}^{M} q_{t,j} \omega_{t,j} \right)$$

$$+ 10B \sqrt{n \left( (\log\log nB^2)_+ + \log(12/\delta) \right)}$$

We invoke Lemma 19 and Lemma 20 with $\delta \to \delta/3$ take a union bound, to bound the variance and $\omega_{t,i}$ terms as well. These lemmas require one application of Theorem 3 to hold simultaneously and no additional union bound is required between them, since the randomness comes only from the confidence interval over $\hat{\theta}_t$.

$$R(n,i) \leq C_1 B n^{3/4} + \frac{\log M}{\eta_n}$$

$$+ C_2 B^2 C_{\min} \sqrt{n} + C_3 B n^{1/4} \left( 1 + \frac{n^{-3/8}}{C_{\min}} \sqrt{\log(Md/\delta) + (\log\log n)_+} \right)$$

$$+ C(M,\delta,d) \frac{\log n}{C_{\min}} \left( 1 + \frac{n^{-3/8}}{C_{\min}} \sqrt{\log(Md/\delta) + (\log\log n)_+} \right)$$

$$+ n^{3/4} C(M,\delta,d) \left( 1 + \frac{n^{-3/8}}{C_{\min}} \sqrt{\log(Md/\delta) + (\log\log n)_+} \right)$$

$$+ 10B \sqrt{n \left( (\log\log nB^2)_+ + \log(12/\delta) \right)}$$

with probability greater than $1 - \delta$, conditioned on event $E$. Assuming that event $E$ happens with probability $1 - 2\delta$, let $\mathcal{B} = \mathcal{B}(\tilde{\delta}, M, d, n, B, \sigma, C_{\min})$ denote the right-hand-side of the regret inequality above.

By the chain rule we may write,

$$\mathbb{P}\big(\text{Reg}(n,i) \le \mathcal{B}\big)$$
$$\ge \mathbb{P}\left(R(n,i) \le \mathcal{B} \Big| \forall t \in [n], j \in [M] : \eta_t \hat{r}_{t,j} \le 1\right) \mathbb{P}\left(\forall t \in [n], j \in [M] : \eta_t \hat{r}_{t,j} \le 1\right)$$
$$\ge \mathbb{P}\left(R(n,i) \le \mathcal{B} \Big| \forall t \in [n], j \in [M] : \eta_t \hat{r}_{t,j} \le 1\right)(1 - 2\delta)$$
$$\ge (1 - \delta)(1 - 2\delta) \ge 1 - 3\delta.$$

It remains to verify that event $E$ is met with probability $1 - 2\delta$. Recall that $\eta_t = \mathcal{O}(C_{\min}/\sqrt{t}C(M,\delta,d))$, and that from Lemma 8 with probability $1 - \delta$,

$$\frac{C_{\min}}{4\sqrt{t}C(M,\delta,d)} \le \frac{C_1 C_{\min} t^{1/4} - C_2 t^{-1/8}\sqrt{\log(Md/\delta) + (\log\log t)_+}}{4\sqrt{10}C(M,\delta,d)} \le \frac{\kappa^2(\Phi_t, 2)}{4\sqrt{10}\lambda_t}$$

Therefore, from Lemma 20, there exists $C_\eta$ such that $\eta_t = C_\eta C_{\min}/B\sqrt{t}C(M,\delta,d)$ satisfying,

$$\mathbb{P}\left(\forall t \ge 1, j \in [M] : \eta_t \hat{r}_{t,j} \le 1\right) \ge 1 - 2\delta$$

The proof is then finished by setting $\delta \leftarrow 3\delta$ (and updating the absolute constants). $\qquad\square$

## D.2 Proof of Virtual Regret

**Proposition 21.** *For any fixed $\tilde{\lambda} > 0$, there exists an absolute constant $C_1$ such that*

$$\mathbb{P}\left(\forall t \ge 1 : \left\|\hat{\boldsymbol{\beta}}_{t,j^\star} - \boldsymbol{\theta}_{j^\star}\right\|_2 \le \omega(t,\delta,d)\right) \ge 1 - \delta.$$

*where*

$$\omega(t,\delta,d) := C_1 \sqrt{\frac{\sigma^2 d \log\left(\frac{t}{\tilde{\lambda}d}+1\right) + 2\sigma^2 \log(1/\delta) + \tilde{\lambda}B^2}{\tilde{\lambda} + C_{\min}t^{3/4}}}\left(1 + C_{\min}^{-1}t^{-3/8}\sqrt{\log(Md/\delta) + (\log\log t)_+}\right).$$

*Moreover, for $u_{t,j^\star}(\cdot) := \hat{\boldsymbol{\beta}}_{t,j^\star}^\top \boldsymbol{\phi}_{j^\star}(\cdot) + \omega(t,\delta,d)$,*

$$\mathbb{P}\left(\forall t \ge 1, \boldsymbol{x} \in \mathcal{X} : r(\boldsymbol{x}) \le u_{t,j^\star}(\boldsymbol{x})\right) \ge 1 - \delta.$$

*Proof of Proposition 21.* Define for convenience $V_t = \Phi_{t,j^\star}^\top \Phi_{t,j^\star} + \tilde{\lambda}\boldsymbol{I}$. We first observe that

$$\hat{\boldsymbol{\beta}}_{t,j^\star} = V_t^{-1}(\Phi_{t,j^\star})^\top \boldsymbol{y}_t$$

We can apply results from Abbasi-Yadkori et al. [2011] to get an anytime-valid confidence set. Their Theorem 2 asserts that with probability $1 - \delta$, for all $t \ge 1$ we have[2]

$$\left\|\hat{\boldsymbol{\beta}}_{t,j^\star} - \boldsymbol{\theta}_{j^\star}\right\|_{V_t}^2 \le \beta_t$$

where

$$\beta_t = 2\sigma^2 \log\left(\frac{\det(V_t)^{1/2}}{\det(\tilde{\lambda}\boldsymbol{I})^{1/2}\delta}\right) + \tilde{\lambda}B^2$$

Clearly, $V_t \succeq \lambda_{\min}(V_t)I$, and therefore with high probability,

$$\left\|\hat{\boldsymbol{\beta}}_{t,j^\star} - \boldsymbol{\theta}_{j^\star}\right\|_2 \le \sqrt{\frac{\beta_t}{\lambda_{\min}(V_t)}}$$

uniformly over time. Our assumption is that $\|\boldsymbol{\phi}_j(\boldsymbol{x})\| \le 1$, and hence, denoting by $\nu_i$ the eigenvalues of $V_t$, the geometric-arithmetic mean inequality yields

$$\det(V_t) \le \prod_{i=1}^d \nu_i \le \left(\frac{1}{d}\text{trace}(V_t)\right)^d.$$

---

[2]Their theorem statement is slightly different, but they prove the stronger version we state below.

Given that

$$\text{trace}(V_t) = \sum_{i=1}^{d}\sum_{s=1}^{t}(\phi_{j^\star}(x))_i^2 + \tilde{\lambda}d \le t + \tilde{\lambda}d$$

we can conclude that

$$\beta_t \le 2\sigma^2 \log\left(\frac{(t/d+\tilde{\lambda})^{d/2}}{\tilde{\lambda}^{d/2}\delta}\right) + \tilde{\lambda}B^2 = d\sigma^2 \log\left(\frac{t}{\tilde{\lambda}d}+1\right) + 2\sigma^2\log(1/\delta) + \tilde{\lambda}B^2$$

We note that

$$\lambda_{\min}(V_t) = \lambda_{\min}(\Phi_{t,j}^\top \Phi_{t,j}) + \tilde{\lambda}.$$

Then due to Lemma 9, there exist absolute constants $C_1$ and $C_2$ such that for all $t \ge 1$,

$$\lambda_{\min}(V_t) \ge \tilde{\lambda} + C_1 C_{\min}t^{3/4} - C_2 t^{3/8}\sqrt{\log(Md/\delta) + (\log\log t)_+}$$

therefore, there exists $C_3$ and $C_4$ such that

$$\frac{1}{\sqrt{\lambda_{\min}(V_t)}} \le \frac{1}{\sqrt{\tilde{\lambda} + C_1 C_{\min}t^{3/4} - C_2 t^{3/8}\sqrt{\log(Md/\delta) + (\log\log t)_+}}}$$

$$\le \frac{C_3}{\sqrt{\tilde{\lambda} + C_1 C_{\min}t^{3/4}}}\left(1 + \frac{t^{3/8}\sqrt{\log(Md/\delta) + (\log\log t)_+}}{\tilde{\lambda} + C_1 C_{\min}t^{3/4}}\right)$$

$$\le \frac{C_4}{\sqrt{\tilde{\lambda} + C_1 C_{\min}t^{3/4}}}\left(1 + C_{\min}^{-1}t^{-3/8}\sqrt{\log(Md/\delta) + (\log\log t)_+}\right)$$

with high probability for all $t \ge 1$. Setting

$$\omega(t,\delta,d) = C_5\sqrt{\frac{\sigma^2 d\log\left(\frac{t}{\tilde{\lambda}d}+1\right) + 2\sigma^2\log(1/\delta) + \tilde{\lambda}B^2}{\tilde{\lambda} + C_{\min}t^{3/4}}}\left(1 + C_{\min}^{-1}t^{-3/8}\sqrt{\log(Md/\delta) + (\log\log t)_+}\right)$$

where $C_5$ is an absolute constant concludes the parametric confidence bound. The upper confidence bound then simply follows: for any $\boldsymbol{x} \in \mathcal{X}$

$$r(\boldsymbol{x}) - \hat{\boldsymbol{\beta}}_{t,j^\star}^\top \boldsymbol{\phi}_{j^\star}(\boldsymbol{x}) = \langle \boldsymbol{\theta}_{j^\star} - \hat{\boldsymbol{\beta}}_{t,j^\star}, \boldsymbol{\phi}_{j^\star}(\boldsymbol{x})\rangle \le \left\|\boldsymbol{\theta}_{j^\star} - \hat{\boldsymbol{\beta}}_{t,j^\star}\right\|_2 \|\boldsymbol{\phi}_{j^\star}(\boldsymbol{x})\|_2 \le \omega(t,\delta,d)$$

where the last inequality holds with high probability simultaneously for all $t \ge 1$. $\qquad\square$

***Proof of Lemma 15.*** Using Proposition 21 and the Cauchy-Schwarz inequality we obtain,

$$\tilde{R}_{j^\star}(n) = \sum_{t=1}^{n} r(\boldsymbol{x}^\star) - r(\tilde{\boldsymbol{x}}_{t,j})$$

$$= \sum_{t=1}^{n} r(\boldsymbol{x}^\star) - \hat{r}_t(\boldsymbol{x}^\star) + \hat{r}_t(\boldsymbol{x}^\star) - \hat{r}_t(\tilde{\boldsymbol{x}}_{t,j}) + \hat{r}_t(\tilde{\boldsymbol{x}}_{t,j}) - r(\tilde{\boldsymbol{x}}_{t,j})$$

$$\le \sum_{t=1}^{n} \left\|\boldsymbol{\theta}_j - \hat{\boldsymbol{\theta}}_{t,j}\right\|_2 \left(\|\boldsymbol{\phi}_j(\boldsymbol{x}^\star)\|_2 + \|\boldsymbol{\phi}_j(\tilde{\boldsymbol{x}}_{t,j})\|_2\right) + \hat{r}_t(\boldsymbol{x}^\star) - \hat{r}_t(\tilde{\boldsymbol{x}}_{t,j})$$

$$\le \sum_{t=1}^{n} \omega(t,\delta,d)\left(\|\boldsymbol{\phi}_j(\boldsymbol{x}^\star)\|_2 + \|\boldsymbol{\phi}_j(\tilde{\boldsymbol{x}}_{t,j})\|_2\right) + \hat{r}_t(\boldsymbol{x}^\star) - \hat{r}_t(\tilde{\boldsymbol{x}}_{t,j})$$

with probability $1 - \delta$. If the agent selects actions greedily, then $\hat{r}_t(\boldsymbol{x}^\star) \le \hat{r}_t(\tilde{\boldsymbol{x}}_{t,j})$, and

$$\tilde{R}_{j^\star}(n) \le \sum_{t=1}^{n} \omega(t,\delta,d)\left(\|\boldsymbol{\phi}_j(\boldsymbol{x}^\star)\|_2 + \|\boldsymbol{\phi}_j(\tilde{\boldsymbol{x}}_{t,j})\|_2\right) \le \sum_{t=1}^{n} 2\omega(t,\delta,d)$$

since the feature map is normalized to satisfy $\|\boldsymbol{\phi}_j(\cdot)\| \le 1$. If the agent selects actions optimistically according to the upper confidence bound of Proposition 21, then

$$\hat{r}_t(\tilde{\boldsymbol{x}}_{t,j}) + \omega(t,\delta,d)\|\boldsymbol{\phi}_j(\tilde{\boldsymbol{x}}_{t,j})\| \ge \hat{r}_t(\boldsymbol{x}^\star) + \omega(t,\delta,d)\|\boldsymbol{\phi}_j(\boldsymbol{x}^\star)\|$$

which implies

$$\hat{r}_t(\boldsymbol{x}^\star) - \hat{r}_t(\tilde{\boldsymbol{x}}_{t,j}) \leq \omega(t,\delta,d)\|\boldsymbol{\phi}_j(\tilde{\boldsymbol{x}}_{t,j})\| - \omega(t,\delta,d)\|\boldsymbol{\phi}_j(\boldsymbol{x}^\star)\|$$

and therefore,

$$\tilde{R}_{j^\star}(n) \leq \sum_{t=1}^n 2\omega(t,\delta,d)\|\boldsymbol{\phi}_j(\tilde{\boldsymbol{x}}_{t,j})\|_2 \leq \sum_{t=1}^n 2\omega(t,\delta,d).$$

Then due to Proposition 21, with probability greater than $1 - \delta$, simultaneously for all $n \geq 1$,

$$\sum_{t=1}^n \omega(t,\delta,d) \leq \sum_{t=1}^n C_1 \sqrt{\frac{\sigma^2 d \log\left(\frac{t}{\tilde{\lambda}d}+1\right) + 2\sigma^2 \log(1/\delta) + \tilde{\lambda}B^2}{\tilde{\lambda} + C_{\min}t^{3/4}}} \left(1 + t^{-3/8}\sqrt{\log(\frac{Md}{\delta}) + (\log\log t)_+}\right)$$

$$\leq \tilde{C}_1 n^{5/8} \sqrt{\frac{\sigma^2 d \log\left(\frac{n}{\tilde{\lambda}d}+1\right) + 2\sigma^2 \log(1/\delta) + \tilde{\lambda}B^2}{C_{\min}}} \left(1 + C_{\min}^{-1}n^{\frac{-3}{8}}\sqrt{\log(\frac{Md}{\delta}) + (\log\log n)_+}\right)$$

concluding the proof. $\qquad\square$

# E  Time-Uniform Concentration Inequalities

We will make use of the elegant concentration results in Howard et al. [2021], which analyzes the boundary of sub-Gamma processes.

**Definition 22** (Sub-Gamma process). Let $(S_t)_{t=0}^\infty$ and $(V_t)_{t=0}^\infty$ be real-valued processes adapted to $(\mathcal{F}_t)_{t=1}^\infty$ with $S_0 = V_0 = 0$ and $V_t$ non-negative. We say that $S_t$ is sub-Gamma if for $\lambda \in [0, 1/c)$, there exists a supermartingale $(M_t(\lambda))_{t=0}^\infty$ w.r.t. $\mathcal{F}_t$, such that $\mathbb{E}\, M_0 = 1$ and for all $t \geq 1$:

$$\exp\{\lambda S_t - \frac{\lambda^2}{2(1-c\lambda)}V_t\} \leq M_t(\lambda) \qquad a.s.$$

The following is a special case of Theorem 1 in Howard et al. [2021]. We have simplified it by making a few straightforward choices for the parameters used originally by Howard et al. [2021], which will yield an easier-to-use bound in our scenario.

**Proposition 23** (Curved Boundary of Sub-Gamma Processes). *Let $(S_t)_{t\geq 0}$ be sub-Gamma with scale parameter $c$ and variance process $(V_t)_{t\geq 0}$. Define the boundary*

$$\mathcal{B}_\alpha(v) := \frac{5}{2}\sqrt{\max\{v,1\}\left((\log\log ev)_+ + \log\left(\frac{2}{\alpha}\right)\right)} + 3c\left((\log\log ev)_+ + \log\left(\frac{2}{\alpha}\right)\right),$$

*for $v > 0$, where $(x)_+ = \max(0,x)$. Then,*

$$\mathbb{P}(\exists t: S_t \geq \mathcal{B}_\alpha(V_t)) \leq \alpha.$$

*Proof of Proposition 23.* Suppose $\xi(\cdot)$ denotes the Riemann zeta function. Theorem 1 in Howard et al. [2021] states that if $(S_t)_{t\geq 0}$ is a sub-Gamma process with variance process $(V_t)_{t\geq 0}$ then the boundary

$$\mathcal{S}_\alpha(v') = k_1\sqrt{v'\left(s\log\log(\eta v') + \log\left(\frac{\zeta(s)}{\alpha\log^s\eta}\right)\right)} + ck_2\left(s\log\log(\eta v') + \log\left(\frac{\zeta(s)}{\alpha\log^s\eta}\right)\right).$$

satisfies,

$$\mathbb{P}(\exists t: S_t \geq \mathcal{S}_\alpha(\max(V_t,1))) \leq \alpha$$

where

$$k_1 := \frac{\eta^{1/4} + \eta^{-1/4}}{\sqrt{2}} \qquad \text{and} \qquad k_2 := (\sqrt{\eta}+1)/2$$

and $s, \eta \geq 1$. Choosing $s = 2$ and $\eta = e$, we obtain $\zeta(2) = \pi^2/6 \leq 2$. Furthermore, we have $k_1 \leq \frac{3}{2}$ and $k_2 \leq \frac{3}{2}$. Then if $v' \geq 1$ (which we will enforce by the construction $v' = \max(1,v)$), we compute

$$s\log\log(\eta v') + \log\left(\frac{\zeta(s)}{\alpha\log^s\eta}\right) \leq 2(\log\log ev')_+ + \log\left(\frac{2}{\alpha}\right).$$

Therefore, we can upper bound (using our bounds on $k_1, k_2$)

$$\mathcal{S}_\alpha(v') \leq \frac{5}{2}\sqrt{v'\left((\log\log ev')_+ + \log\left(\frac{2}{\alpha}\right)\right)} + 3c\left((\log\log ev')_+ + \log\left(\frac{2}{\alpha}\right)\right).$$

Now, since the boundary is given by $\mathcal{S}_\alpha(\max(v,1))$ and $v' = \max(v,1) \geq 1$ we deduce that

$$\mathcal{B}_\alpha(v) := \frac{5}{2}\sqrt{\max\{v,1\}\left((\log\log ev)_+ + \log\left(\frac{2}{\alpha}\right)\right)} + 3c\left((\log\log ev)_+ + \log\left(\frac{2}{\alpha}\right)\right).$$

is an any-time valid boundary. $\qquad\square$

**Lemma 24** (Time-Uniform Two-sided Bernoulli). *Let $X_1, \ldots, X_s, \ldots, X_t$ be a martingale sequence of Bernoulli random variables with conditional mean $\gamma_s$. Then for all $\delta > 0$,*

$$\mathbb{P}\left(\exists t : \left|\sum_{s=1}^{t}(X_s - \gamma_s)\right| \geq \frac{5}{2}\sqrt{t\left((\log\log t)_+ + \log(4/\delta)\right)}\right) \leq \delta,$$

*Proof of Lemma 24.* By Proposition 23, we know that if $S_t$ is sub-Gamma with variance process $V_t$ and scale parameter $c$, then

$$\mathbb{P}\left(\exists t : S_t \geq \mathcal{B}_\delta(V_t)\right) \leq \delta,$$

where

$$\mathcal{B}_\delta(v) := \frac{5}{2}\sqrt{\max\{1,v\}\left((\log\log ev)_+ + \log(2/\delta)\right)} + 3c\left((\log\log ev)_+ + \log(2/\delta)\right).$$

By Howard et al. [2020], we know that if $(X_t)_{t=1}^{\infty}$ is a Bernoulli sequence, then $S_t = \sum_{s=1}^{t}(X_s - \gamma_s)$ is sub-Gamma with variance process $V_t = t$ and scale parameter $c = 0$ (hence, sub-Gaussian). This implies,

$$\mathbb{P}\left(\exists t : \sum_{s=1}^{t}(X_s - \gamma_s) \geq \frac{5}{2}\sqrt{t\left((\log\log t)_+ + \log(2/\delta)\right)}\right) \leq \delta,$$

The above arguments also holds for the sequence $Z_s = -X_s$. Then taking a union bound and adjusting $\delta \leftarrow \delta/2$ concludes the proof. $\qquad\square$

**Lemma 25** (Time-Uniform Bernstein). *Let $(\xi_i)_{i=1}^{\infty}$ be a sequence of conditionally standard sub-gaussian variables, where each $\xi_i$ is $\mathcal{F}_{i-1} = \sigma(\xi_1, \ldots, \xi_i)$ measurable. Then, for $v_i \in \mathbb{R}$ and $\delta \in (0,1]$*

$$\mathbb{P}\left(\exists t : \sum_{i=1}^{t}(\xi_i^2 - 1)v_i \geq \frac{5}{2}\sqrt{\max\left\{1, 4\|\boldsymbol{v}_t\|_2^2\right\}\omega_\delta(\|\boldsymbol{v}_t\|_2)} + 12\omega_\delta(\|\boldsymbol{v}_t\|_2)\max_{i\geq 1} v_i\right) \leq \delta$$

*where, $\boldsymbol{v}_t = (v_1, \ldots, v_t) \in \mathbb{R}^t$ and $\omega_\delta(v) := \left(\log\log(4ev^2)\right)_+ + \log(2/\delta)$.*

*Proof of Lemma 25.* From Lemma 4, $S_t = \sum_{i=1}^{t}(\xi_i^2 - 1)v_i$ is sub-Gamma with variance process $V_t = 4\sum_{i=1}^{t}v_i^2$ and $c = 4\max_{i\geq 1} v_i$. By Proposition 23, we know that if $S_t$ is sub-Gamma with variance process $V_t$ and scale parameter $c$, then

$$\mathbb{P}\left(\exists t : S_t \geq \mathcal{B}_\delta(V_t)\right) \leq \delta,$$

where

$$\mathcal{B}_\delta(v) := \frac{5}{2}\sqrt{\max\{1,v\}\left((\log\log ev)_+ + \log(2/\delta)\right)} + 3c\left((\log\log ev)_+ + \log(2/\delta)\right).$$

$\qquad\square$

**Lemma 26** (Time-Uniform Azuma-Hoeffding). *Let $X_1, \ldots, X_n$ be a martingale difference sequence such that $|X_t| \leq B$ for all $t > 1$ almost surely. Then for all $\delta > 0$,*

$$\mathbb{P}\left(\exists t : \sum_{s=1}^{t}X_s \geq \frac{5B}{2}\sqrt{t\left((\log\log etB^2)_+ + \log(2/\delta)\right)}\right) \leq \delta,$$

*Proof of Lemma 26.* By Proposition 23, we know that if $S_t$ is sub-Gamma with variance process $V_t$ and scale parameter $c$, then

$$\mathbb{P}\left(\exists t: \ S_t \geq \mathcal{B}_\delta(V_t)\right) \leq \delta,$$

where

$$\mathcal{B}_\delta(v) := \frac{5}{2}\sqrt{\max\{1,v\}\left((\log\log ev)_+ + \log(2/\delta)\right)} + 3c\left((\log\log ev)_+ + \log(2/\delta)\right).$$

By [Howard et al., 2020], we know that if $(X_t)_{t=1}^\infty$ is $B$-bounded martingale difference sequence, then $S_t = \sum_{s=1}^t X_s$ is sub-Gamma with variance process $V_t = tB^2$ and scale parameter $c = 0$. This implies,

$$\mathbb{P}\left(\exists t: \ S_t \geq \frac{5B}{2}\sqrt{t\left((\log\log etB^2)_+ + \log(2/\delta)\right)}\right) \leq \delta,$$

concluding the proof. □

## F    Experiment Details

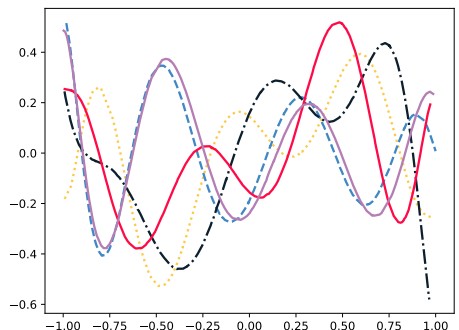

Figure 5: Examples of possible reward functions $r(\cdot)$ in our experiments.

### F.1    Hyper-Parameter Tuning Results

We implement 6 algorithms in our experiments, ETC [Algorithm 4, Hao et al., 2020], ETS (Algorithm 5), CORRAL [Algorithm 6, Agarwal et al., 2017], ALEXP (Algorithm 1), and Lastly UCB (Algorithm 3) with the oracle feature map $\phi_{j\star}$ (Oracle), and UCB with the concatenated feature map $\phi$ (Naive). The Python code is available on github.com/lasgroup/ALEXP. When algorithms require exploration, e.g., in the case of ETC or ALEXP, we simply set $\pi = \text{Unif}(\mathcal{X})$. Figure 7 shows the results of our hyperparameter tuning experiment. To ensure that the curves are valid, we run each

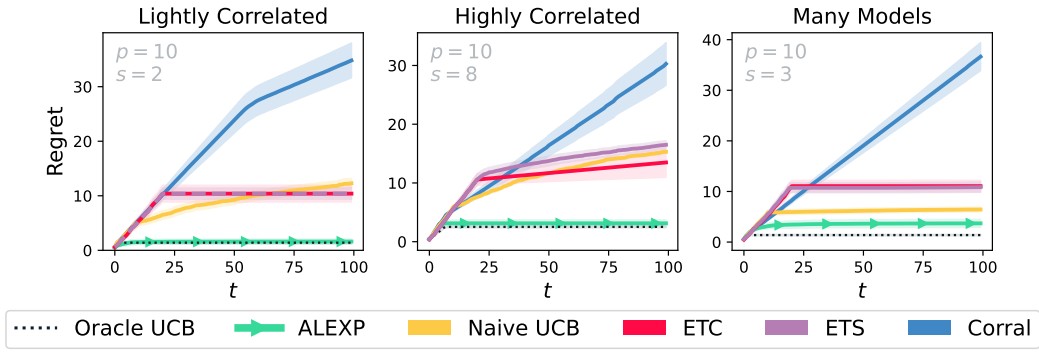

Figure 6: Bench-marking ALEXP and other baselines. Complete version of Fig. 1 and Fig. 2.

---

**Algorithm 2** GetPosterior

Inputs: $H_t, \phi, \tilde{\lambda}$
Let $K_t \leftarrow [\phi^\top(\boldsymbol{x}_i)\phi(\boldsymbol{x}_j)]_{i,j \le t}$, and $V_t \leftarrow (K_t + \tilde{\lambda}^2 \boldsymbol{I})$, and $\boldsymbol{k}(\cdot) \leftarrow [\phi^\top(\boldsymbol{x}_i)\phi(\cdot)]_{i \le t}$

Calculate $\mu_t(\cdot) \leftarrow \boldsymbol{k}^T(\cdot)V_t^{-1}\boldsymbol{y}_t$

Calculate $\sigma_t(\cdot) \leftarrow \sqrt{\phi^\top(\cdot)\phi(\cdot) - \boldsymbol{k}^\top(\cdot)V_t^{-1}\boldsymbol{k}(\cdot)}$
Return: $\mu_t, \sigma_t$

---

**Algorithm 3** UCB

Inputs: $\tilde{\lambda}, \beta_t, \phi$
**for** $t = 1, \ldots, n$ **do**
    Choose $\boldsymbol{x}_t \arg\max u_{t-1}(\boldsymbol{x}) = \mu_{t-1}(\boldsymbol{x}) + \beta_t\sigma_{t-1}(\boldsymbol{x})$.      ▷ Choose actions optimistically
    Observe $y_t = r(\boldsymbol{x}_t) + \varepsilon_t$.      ▷ Receive reward
    $H_t \leftarrow H_{t-1} \cup \{(\boldsymbol{x}_t, y_t)\}$      ▷ Append history
    Update $\mu_t, \sigma_t \leftarrow \text{GetPosterior}(H_t, \phi, \tilde{\lambda})$
**end for**

---

configuration for 20 different random seeds, i.e. on different random environments. The shaded areas in Figure 7 show the standard error.

**UCB.** For all the experiments, we set the exploration coefficient of UCB to $\beta_t = 2$[3] and choose the regression regularizer from $\tilde{\lambda} \in \{0.01, 0.1, 0.5\}$. We use PYTORCH [Paszke et al., 2017] for updating the upper confidence bounds, which requires more regularization for longer feature maps (e.g. when $s = 8$, $p = 2$), to be computationally stable.

**Lasso.** Every time we need to solve Eq. (1), we set $\lambda_t$ according to the *rate* suggested by Theorem 3. To find a suitable constant scaling coefficient, we perform a hyper-parameter tuning experiment sampling 20 values in $[10^{-5}, 10^0]$. We choose $\lambda_0 = 0.009$, and scale $\lambda_t$ with it across all experiments.

**ALEXP.** We set the rates for $\gamma_t$ and $\eta_t$ as prescribed by Theorem 1. For the scaling constants, we perform a hyper-parameter tuning experiment log-uniformly sampling 20 different configurations from $\gamma_0 \in [10^{-4}, 10^{-1}]$ and $\eta_0 \in [10^0, 10^2]$. For each problem instance (i.e. as $s$ and $p$ change) we repeat this process. However we observe that the optimal hyper-parameters work well across all problem instances.

**ETC/ETS.** For these algorithms, we separately tune $n_0$ for each problem instance. We set $\lambda_1 \propto \sqrt{\log M / n_0}$ according to Theorem 4.2 of [Hao et al., 2020] and scale it with $\lambda_0 = 0.009$, as stated before. We uniformly sample 10 different values where $n_0 \in [2, 80]$ since the horizon is $n = 100$. The optimal value often happens around $n_0 = 20$.

**CORRAL.** We set the rates of the parameters as $\gamma = \mathcal{O}(1/n)$ and $\eta = \mathcal{O}(\sqrt{M/n})$ according to Agarwal et al. [2017, Theorem 5,]. Then similar to ALEXP, we tune the scaling constants. The procedure for tuning the constants is identical to ALEXP, as in we use the same search interval, and try 10 different configurations for $\gamma$ and $\eta$.

---

[3]To achieve the $\sqrt{dT \log T}$ regret, one has to set $\beta_t = \mathcal{O}(\sqrt{d \log T})$ as shown in Proposition 21.

**Algorithm 4** ETC [Hao et al., 2020]

---

Inputs: $n_0, n, \lambda_1, \pi$
Let $H_0 = \emptyset$
**for** $t = 1, \ldots, n_0$ **do**
    Draw $\boldsymbol{x}_t \sim \pi$.                                            ▷ Explore.
    Observe $y_t = r(\boldsymbol{x}_t) + \varepsilon_t$.                            ▷ Receive reward
    $H_t \leftarrow H_{t-1} \cup \{(\boldsymbol{x}_t, y_t)\}$                        ▷ Append history
**end for**
$\hat{\boldsymbol{\theta}}_{n_0} \leftarrow \mathcal{L}(\boldsymbol{\theta}, H_{n_0}, \lambda_1)$                               ▷ Perform Lasso once
**for** $t = n_0 + 1, \ldots, n$ **do**
    Choose $\boldsymbol{x}_t = \arg\max \hat{\boldsymbol{\theta}}_{n_0}^\top \boldsymbol{\phi}(\boldsymbol{x})$                ▷ Choose actions greedily
**end for**

---

**Algorithm 5** ETS

---

Inputs: $n_0, n, \lambda_1, \tilde{\lambda}, \beta_t, \pi$
Let $H_0 = \emptyset$
**for** $t = 1, \ldots, n_0$ **do**
    Draw $\boldsymbol{x}_t \sim \pi$.                                            ▷ Explore
    Observe $y_t = r(\boldsymbol{x}_t) + \varepsilon_t$.                            ▷ Receive reward
    $H_t \leftarrow H_{t-1} \cup \{(\boldsymbol{x}_t, y_t)\}$                        ▷ Append history
**end for**
$\hat{\boldsymbol{\theta}}_{n_0} \leftarrow \mathcal{L}(\boldsymbol{\theta}, H_{n_0}, \lambda_1)$                               ▷ Perform Lasso once
$\hat{J} \leftarrow \{j \mid \hat{\boldsymbol{\theta}}_{n_0,j} \neq \boldsymbol{0}, j \in [M]\}$                   ▷ Get sparsity pattern
$\boldsymbol{\phi}_{\hat{J}}(\cdot) \leftarrow [\boldsymbol{\phi}_j(\cdot)]_{j \in \hat{J}}$                          ▷ Model-select acc. to $\hat{J}$
**for** $t = n_0 + 1, \ldots, n$ **do**
    Choose $\boldsymbol{x}_t = \arg\max u_{t-1}(\boldsymbol{x}) = \mu_{t-1}(\boldsymbol{x}) + \beta_t \sigma_{t-1}(\boldsymbol{x})$     ▷ Choose actions optimistically
    Observe $y_t = r(\boldsymbol{x}_t) + \varepsilon_t$
    $H_t \leftarrow H_{t-1} \cup \{(\boldsymbol{x}_t, y_t)\}$
    Update $\mu_t, \sigma_t \leftarrow \text{GetPosterior}(H_t, \boldsymbol{\phi}_{\hat{J}}, \tilde{\lambda})$
**end for**

---

**Algorithm 6** CORRAL [Agarwal et al., 2017]

---

Inputs: $n, \gamma, \eta$
Initialize $\beta = e^{1/\ln n}$, $\eta_{1,j} = \eta$, $\rho_{1,j} = 2M$ for all $j = 1, \ldots, M$
Set $\boldsymbol{q}_1 = \bar{\boldsymbol{q}}_1 = \frac{1}{M}$ and initialize base agents $(p_{1,1}, \ldots, p_{1,M})$.
**for** $t = 1, \ldots, n$ **do**
    Choose $j_t \sim \bar{\boldsymbol{q}}_t$.                                      ▷ Sample Agent
    Draw $\boldsymbol{x}_t \sim p_{t,j_t}$.                           ▷ Play action according to agent $j_t$
    Observe $y_t = r(\boldsymbol{x}_t) + \varepsilon_t$.
    Calculate IW estimates $\hat{r}_{t,j} = \frac{y_t}{\bar{q}_{t,j}} \mathbb{I}\{j = j_t\}$ for all $j = 1, \ldots, M$.
    Send $\hat{r}_{t,j} = \frac{y_t}{\bar{q}_{t,j}} \mathbb{I}\{j = j_t\}$ to agents and get updated policies $p_{t+1,j}$.
    $\boldsymbol{q}_{t+1} = \text{LOG-BARRIER-OMD}(\boldsymbol{q}_t, \hat{r}_{t,j_t} \boldsymbol{e}_{j_t}, \boldsymbol{\eta}_t)$        ▷ Update agent probabilities
    $\bar{\boldsymbol{q}}_{t+1} = (1 - \gamma)\boldsymbol{q}_{t+1} + \gamma \frac{1}{M}$            ▷ Mix with exploratory distribution
    **for** $j = 1, \ldots, M$ **do**                            ▷ Update parameters
        **if** $\frac{1}{\bar{q}_{t+1,j}} > \rho_{t,j}$ **then** $\rho_{t+1,j} \leftarrow \frac{2}{\bar{q}_{t,j}}$, and $\eta_{t+1,j} \leftarrow \beta \eta_{t,j}$
        **else** $\rho_{t+1,j} \leftarrow \rho_{t,j}$ and $\eta_{t+1,j} \leftarrow \eta_{t,j}$
        **end if**
    **end for**
**end for**

---

**Algorithm 7** LOG-BARRIER-OMD

---

Inputs: $\boldsymbol{q}_t, \boldsymbol{\ell}_t, \boldsymbol{\eta}_t$
Find $\xi \in [\min_j \ell_{t,j}, \max_j \ell_{t,j}]$ such that $\sum_{j=1}^M \left(q_{t,j}^{-1} + \eta_{t,j}(\ell_{t,j} - \xi)\right)^{-1} = 1$
Return: $\boldsymbol{q}_{t+1}$ where $q_{t+1,j}^{-1} = q_{t,j}^{-1} + \eta_{t,j}(\ell_{t,j} - \xi)$ for all $j \in [M]$

---

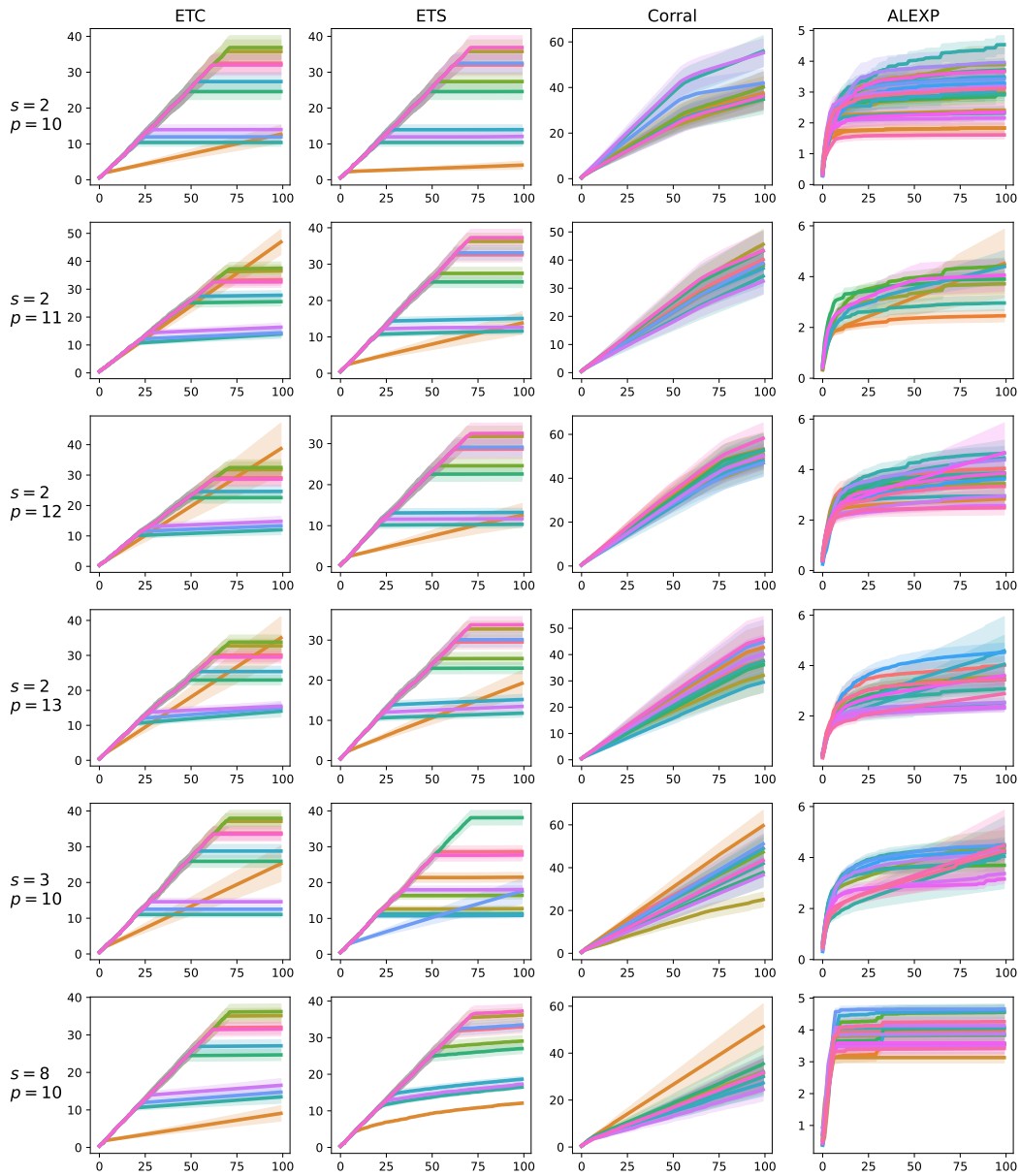

Figure 7: Results for different hyper-parameters across different problem instances. ALEXP is robust to the choice of hyper-paramters.

