# OpenReview forum: "Anytime Model Selection in Linear Bandits"
_NeurIPS.cc/2023/Conference — NeurIPS 2023 poster_

### Official Review · Reviewer_XdEK · 2023-07-04

**Soundness:** 3 good
**Presentation:** 3 good
**Contribution:** 3 good
**Rating:** 6
**Confidence:** 3

**Summary:**

This paper introduces AlExp, a novel algorithm to address the problem of online model selection in the context of bandit optimization. The algorithm interacts with the environment by randomly choosing a learner (linear bandit regret minimizer) each round, then the chosen learner tries to propose the optimal action. The exploration-exploitation trade-off is dealt with both by randomizing the choice of the learner and by the learner's policy itself. The authors provide theoretical guarantees and empirical validation and compare this approach with the existing literature.


**Strengths:**

+ The paper presents an extensive literature review and a detailed comparison between the proposed approach and the existing ones.

+ The algorithm's routine is easy to understand, and its logic is well-commented.

+ The algorithm effectively improves in some dependencies (e.g., the number of learners) w.r.t. to the existing approaches, both theoretically and empirically.

+ The analysis presents technical novelty.


**Weaknesses:**

- In some points, the presentation is not easy to follow, and it will be nice to have a more intuitive grasp of the theoretical quantities considered.

- It will be nice to have more discussion on the computational aspects of the proposed approach: numerically computing some of the quantities seems to be very expensive and prohibitive when the number of learners scales. This contrasts with the main strength of the approach, i.e., the better regret dependency on the number of learners.

- The algorithm requires the knowledge of theoretical quantities (e.g., expected values). In practice, this can be bypassed by a Monte Carlo sampling procedure. However, when performing finite sampling, there's the need to explain how quickly the estimator converges to the theoretical quantity and explain the computational consequences (see previous point) or eventual corrections to the algorithm to deal with the estimation error (e.g., adding an upper confidence bound on the quantities) if due to computational reasons only a small number of samples can be generated.

**Questions:**

+ Can you please provide an examination of the computational limitations of this approach? E.g., providing the solving complexity of the optimizations involved.

+ Due to the strong dependence (on both algorithm and analysis) on quantity C, can you please provide a more intuitive grasp on the magnitude of this quantity w.r.t. other problem-dependent quantities? Is there an available lower/upper bound on it? I feel that the quality of your theoretical results strongly depends on this quantity, and a characterization (at least qualitative) of its magnitude should be discussed.


**Limitations:**

- I feel that the main limitation of this work relies on its application in practice. However, the authors performed a sufficient experimental campaign, validating their approach. Unfortunately, the code for the experimental campaign has not been released, so reproducibility is another limitation of this work (even if they provide details to reproduce the experiments).

---

> ### Author Rebuttal · Authors · 2023-08-09
>
> Thank you for your review. Our response to your questions and concerns follows.
>
>
> **"The algorithm has a strong dependency on quantity $C$, can you provide intuition on it?"** Could you please clarify which $C$ are you referring to? In the text we have $C_{\mathrm{min}}$ and $C(M, d, \delta)$. We give a response on both. We have also updated the paper to give intuition on how $C_{\mathrm{min}}$ affects the algorithm.
>
> - **On $C_{\mathrm{min}}$:** If the action domain is *explorable* then $C_{\mathrm{min}}$ is large, and in this case our bounds improve.
> In many scenarios, this quantity can be treated as an absolute constant, e.g. 1, since we normalize the actions and the feature maps. In the revised text, *we have added a new appendix section* which presents *lower bounds on this quantity* under two scenarios. We prove that if the action domain is a convex body, or if the feature maps are orthonormal, this quantity is an absolute constant. *We have also added a corollary* to the main theorem, which bounds the regret for orthogonal feature map, **independent** of $C_{\mathrm{min}}$, and with the same rate as the main theorem.
>
> - **On $C(M, d, \delta)$:** This quantity has no intrinsic meaning and is defined only for a compact presentation of the theorem. The performance of the algorithm is not tied to it, rather, it depends on the values of the parameters $M, n, d, \delta$ and $C_{\mathrm{min}}$. Regarding its value, as defined in the paper
> $$C(M, \delta, d) =  C_1\sigma\sqrt{1 +  \log(M/\delta) + (\log\log d)_+  +\sqrt{\log (M/\delta)+(\log\log d)}}$$
> where $C_1$ is an absolute constant larger than $160\sqrt{10}$.
>
> **"The Algorithm has to calculate theoretical quantities, e.g. expected values, which are costly."** Many BO agents (e.g. UCB or Greedy) take actions deterministically, therefore the probability distribution $p_{t,j}$ is a single dirac delta at some point $\boldsymbol x_{t,j}$, which for instance maximizes the UCB. Taking the expectation w.r.t  $p_{t,j}$, boils down to simply evaluating the function $\hat{\boldsymbol\theta}^\top_t\boldsymbol \phi(\cdot)$ at the point $\boldsymbol x_{t,j}$, and will not need sampling techniques. As you mentioned, upon using complex randomized agents, the expectations need to be approximated.
>
>
>
> **On computational complexity of ALEXP.**
> The computational complexity of the algorithm scales linearly with $M$, since all agents/models have to be updated. However, updating the models can be done fully in parallel (using tools such as Ray), in which case, increasing the number of models will not affect the runtime of the algorithm. Even without parallelization, the algorithm is light and one complete run of ALEXP with $n=100$, $d=2$, and $M=45$ takes $2.9\pm0.2$ minutes on a single CPU core. Figure 1 in the rebuttal pdf supplement shows how the run-time of our algorithm and other baselines scale with $M$.
>
>
> In general, we suspect that without further assumptions (e.g. relation between $\boldsymbol \phi_j$) linear computational dependency on M may not be avoidable. While algorithms such as CORRAL update only one of the models at every step $t$, their statistical complexity scales polynomially with $M$ (since they require more steps to converge), which results in an overall polynomial computational complexity with $M$.
>
> In the revised paper, we have added Figure 1, and a discussion on computational complexity to Appendix F on “Experiment Details”.
> This being said, We would like to highlight that the contributions of this paper are primarily theoretical. We provide experiments mainly to give additional insights.
>
>
> **"Reproducibility is limited due to lack of code."** We will publicly release the code upon acceptance. In general, the implementations are fairly standard and straightforward (cf. pages 31-34 for details).

---

> > ### Comment · Reviewer_XdEK · 2023-08-21
> >
> > Thank you for your response. The authors addressed my concerns, except for the ones on computational complexity. I will keep my score for now.

---

> > > ### Author Response · Authors · 2023-08-22
> > >
> > > Thank you for your reply. Is it possible that you have perhaps missed the paragraph titled "On Computational complexity of ALEXP" in our rebuttal?
> > >
> > > In your review you had asked for "an examination of the computational limitations of this approach". As a response, in our rebuttal we provided runtime curves of our algorithm and the oracle. Moreover, we conjectured that model selection algorithms will require $O(M)$ many operations to converge. We believe this can not be lifted, unless certain correlation/structure is assumed between models.
> > > This being said, we would like to highlight that our work is primarily of theoretical nature, and is the first to show that sample-efficient model selection is possible on generic action domains in linear setting.

---

### Official Review · Reviewer_XArQ · 2023-07-06

**Soundness:** 3 good
**Presentation:** 3 good
**Contribution:** 2 fair
**Rating:** 6
**Confidence:** 5

**Summary:**

based on the time-uniform analysis of the Lasso, the anytime exponential weighting algorithm based on Lasso reward estimates with the nature of anytime regret guarantees on model selection linear bandits is developed. The result neither requires knowledge of the horizon n, nor relies on an initial purely exploratory stage.

**Strengths:**

the anytime exponential weighting algorithm based on Lasso reward estimates are horizon-independent, explores adaptively without requiring an initial exploration stage.

**Weaknesses:**

1 The dimensionality of the proposed algorithm model will be ultra-high, and the complexity of the algorithm's runtime depends on the efficiency of the sparse regression model algorithm.
2 The results of the algorithm still belong to the class of multiplicative weights Update algorithms.

**Questions:**

1 What is the relationship between the number of models and their final performance?
2 During the entire learning process, for two cases, i.e., the number of models is fixed, and the number of models is changed. Can you consider these situations and discuss them?

**Limitations:**

the authors adequately addressed the limitations.

---

> ### Author Rebuttal · Authors · 2023-08-09
>
> Thank you for your review.
>
> **Our Contributions.** We would like to highlight the contributions of this paper as it seems to have missed the attention of the reviewer.
> We address the open problem of Agarwal et al. 2017, and are the *first* to show the feasibility of the conjectured $\log M$ rate for model selection on infinite action domains, when the reward is linearly parameterizable. This is a theoretically challenging problem, and our work is the first to show that such rates are attainable.
> Our algorithm ALEXP demonstrates how one can perform adaptive model selection while simultaneously optimizing for an objective, at a $\log M$ rate. Our experiments show that It performs on par with an oracle solver, which knowledge of the true model.
> Crucially, this work presents a *novel time-uniform analysis of the Lasso* and establishes an important connection between online learning and high-dimensional statistics, as pointed our by reviewer Vj2P.
>
>
> **Questions.**
> 1. What is the relationship between the number of the models and their performance?
>
>  There is no relationship. The only assumption is that there exists one model that is able to solve the problem.
>
> 2. During the entire learning process, for two cases, i.e., the number of models is fixed, and the number of models is changed. Can you consider these situations and discuss them?
>
> Following prior work on model selection, we assume that the number of models is fixed during the learning process. A problem setting where new models are introduced during learning could be an interesting future avenue of research.
>
> **Weaknesses.**
> 1. "The dimensionality of the proposed algorithm model will be ultra-high.”
>
> Could you please clarify what you mean by dimensionality of the algorithm? ALEXP can be computationally expensive, as it simultaneously updates all $M$ agents. However, this step can be fully parallelized with tools such as Ray. This way, the runtime of the algorithm will remain constant as $M$ grows.
>
> 2. “The algorithm belongs to the class of multiplicative weights algorithms.”
>
> Indeed exponential weights is a variant of the multiplicative weights algorithm. Could you please clarify why you find this to be a weakness?
>
> Having addressed the points you raised in your review, we kindly ask you to reconsider your assessment of our paper. We would appreciate it if you further express your questions and concerns, particularly given that you found the contributions of this work to be limited. We would be happy to answer them.

---

> > ### Comment · Reviewer_Vj2P · 2023-08-10
> >
> > Since the authors are too polite to put this plainly, I will do so: this reviewer put down a confidence level of 5, yet their review is nonsensical. I would urge the AC to disregard this review completely.

---

### Official Review · Reviewer_Vj2P · 2023-07-06

**Soundness:** 4 excellent
**Presentation:** 4 excellent
**Contribution:** 4 excellent
**Rating:** 8
**Confidence:** 2

**Summary:**

The paper uses tackles model selection for linear bandits with $M$ models. In particular, rewards are estimated from the $M$ models using Lasso and then EXP4 is ran on-top of these estimated rewards to update individual model probabilities. The use of Lasso over ridge regression reduces variance, leading to a $\log M$ dependence rather than usual $\text{rm}poly \, M$. The paper extends the usual martingale mixture analysis from ridge to lasso; this is something I thought would be quite tricky. I look forward to reading it in more detail in the future.

**Strengths:**

Model selection is a very important problem. Problem is explained and motivated very well. Paper is very well written and polished to the standard of a camera-ready.

**Weaknesses:**

I'm not an expert on model selection in linear bandits or full information learning. As such, I cannot identify any weaknesses in this work.

It took me a couple minutes to see why (1) is indeed (group) Lasso---I was expecting to see a 1-norm. I see that you have a sentence or two explaining this after, but somehow this didn't do the trick for me. Maybe some rewording or an extra comment here could be useful, relating it to the more usual notion of Lasso.

**Questions:**

None

**Limitations:**

Limitations clearly discussed in text.

---

> ### Author Rebuttal · Authors · 2023-08-09
>
> Thank you for your review.  Our response follows.
>
> **Add more insight on the group Lasso loss.** Thank you for pointing this out. In the revised version, we have included an explanation of the loss, focusing on how the $(2-1)$-norm induces sparsity at the group level.
>
> **Extending online Ridge analysis to Lasso is a tricky problem.** This is indeed a precise observation! We present time-uniform confidence sets, which shrink at *the lasso fast rate*. This is the key technical contribution which allows us to solve the model selection problem.

---

> > ### Comment · Reviewer_Vj2P · 2023-08-10
> >
> > It seemed to me that this paper would be a done deal in terms of acceptance, and hence I have not bothered to write a lengthy review.
> >
> > This appears to have been a mistake. I will try to find the time to provide a more in-depth review shortly.

---

### Official Review · Reviewer_1XGh · 2023-07-12

**Soundness:** 4 excellent
**Presentation:** 3 good
**Contribution:** 3 good
**Rating:** 7
**Confidence:** 4

**Summary:**

The paper considers the problem of model selection in (lifted) linear bandits. There are M hypothesis models, each of which have a different feature map, and one of these is the true model; it is unknown to the optimizer which of the M models is the correct one. At each timestep, the optimizer chooses one model and obtains instantaneous regret in accordance with the chosen model's selection. The goal is to minimize overall regret across n timesteps.

If feedback were obtained for all model's chosen query at each timestep, then the multiplicative weights update could be used to update the probability of any model being the true oracle model. And the guarantees from the literature would apply. However, we have limited information and only observe the reward for the chosen model's query.

To get around this obstacle, the authors consider an aggregated model with all M model's features combined together. And they train this aggregated model on all the data using a sparse LASSO estimator. Then, they use this aggregated model to fill in the missing regrets for the unchosen models. And these imputed values are then used for the MW updates.

The analysis consists of proving the recovery result on the LASSO, and adjusting the original proof to use the imputed data instead of the original. The authors verify their findings on synthetic data.

**Strengths:**

Clarity:
The problem is well presented, and the key difficulties of the problem are clearly identified - namely the missing regret data for unchosen models. The exposition of the solution is geared towards addressing the difficulty by imputing the missing values.

The structure of the proof is well-outlined in the main text. I also appreciated the intuitive explanation around introducing bias to reduce overall error as the reasoning behind the success of the algorithm.

Quality:
The arguments appear correct and the correct tools are used to obtain the desired results.

Significance and Originality:

As far as I know, this problem has not been considered before, and the improvement in the scaling with M is helpful.

**Weaknesses:**

My main concern is Q1 below.

Other weaknesses are mostly minor

1. It would be useful for the authors to highlight more real-world examples where bandits with multiple models would be most applicable.
2. In line with the above, real-data experiments would be useful to verify the applicability of the algorithm. However, the theoretical contributions are already sufficient so I think this can be deferred to future work.
3. Figure 3 is missing label on the vertical axis and is difficult to interpret.

**Questions:**

1. If the LASSO performs well already, then what is the need to use the LASSO to perform model selection. Why can we not just use the LASSO directly?

2. While the rewards for the query point of the unchosen model are not available, the predictions on the chosen $x_t$ are available. So, why can we not use the prediction error on these points for all the models to choose the weights? Instead of using the imputed data.

3. What is special about the LASSO that makes it suitable for the choice of the aggregate model?

**Limitations:**

Yes

---

> ### Author Rebuttal · Authors · 2023-08-09
>
> Thank you for your review! We have updated the Introduction and Experiments section, incorporating your feedback. Regarding your questions, see below.
>
> **What is special about Lasso that makes it a suitable choice?** To obtain $\log M$ rates in online model selection, we require a reward estimator/hallucinator, whose bias and variance *both* scale with $\mathcal{O}(\log M)$. Lasso happens to hit this balance, while OLS, Ridge, or Importance Weighted estimates have a variance which grows with $\sqrt{M}$.
>
> **When updating the models, why not hallucinate the reward for the chosen actions?** We actually considered this, but did not look into it too far, as it seemed not to affect the rate of the regret bound. We expect this approach to also work, and would be interesting to see how it changes the overall dynamics.
>
> **Why not just use LASSO directly?** Our randomized algorithm is more robust, particularly when features are correlated. We maintain a probability distribution based on Lasso estimates that encourages exploration on model selection level. In contrast, Lasso will deterministically discard some of the models, and in cases (e.g. orthogonal feature maps) will *never* sample them again.
>
> In practice, Lasso often does not perform well for variable selection.The variable selection property of lasso highly relies on orthogonality of the feature maps, and the choice of regularization parameter. In our experiments, we introduce the ETS algorithm (cf. Algorithm 5), which uses Lasso for variable selection to select the model, and we see how it fails when there are many models, or the models are correlated (cf. Figure 1 in the paper).
>
> On the technical side, our analysis allows for bounding the model selection regret, i.e. directly comparing the reward obtained by ALEXP with the oracle agent. It is not clear to us if/how this type of guarantee would be possible when performing Lasso variable selection to select the model.

---

> > ### Comment · Reviewer_1XGh · 2023-08-21
> > **Thanks for response**
> >
> > I thank the authors for their response. I will keep my score.

---

### Official Review · Reviewer_XgJC · 2023-07-26

**Soundness:** 3 good
**Presentation:** 3 good
**Contribution:** 3 good
**Rating:** 6
**Confidence:** 2

**Summary:**

This paper considers linear bandit problem given $M$ models, or sets of feature mappings.
In this problem, it is necessary to select the appropriate action as well as the model based on the bandit feedback.
This paper provides an algorithm with an anytime regret bound of $O(n^{3/4} \sqrt{\log M} + \sqrt{n \log^3 M} + n^{5/8}\sqrt{d \log n})$,
where $n$ and $d$ represent the time horizon and the dimensionality of each model.

**Strengths:**

- The motivation for the research is well explained.
- Experimental results support the effectiveness of the proposed method.

**Weaknesses:**

- The obtained regret bound includes $O(n^{3/4})$-term, which means that the bound is suboptimal if the number of rounds is large.
- The proposed algorithm requires $O(M)$-time computation in each round. This can be a major computational bottleneck. In fact, in the application of sparse linear bandits, $|M|=\binom{p}{s}$, which is exponentially large w.r.t. $s$.
Further, we need $M = \Omega(\exp(n^{1/4}))$ to ensure the regret is of $O(n \log^3 M)$ (i.e., to ensure $n^{3/4} \sqrt{\log M}< \sqrt{n \log^3 M}$).
This means that the proposed algorithm is effective only when $M$ is exponentially large w.r.t. $n$.

**Questions:**

- Can you add a description of the known regret lower bound and how it compares to the main result?
- Line 290: $\{1, ..., p \}$ <- $\{ 0, 1, ..., p \}$ ?
- Line 274-275: "the rate conjectured by Agarwal et al. [2017]" Can you tell me where I can find the corresponding description in Agarwal et al. [2017]?
- The condition of $\eta_t = O(..)$ and $\gamma_t = O(..)$ in Theorem 1: Do the authors mean $\Theta(..)$? I guess that too small $\eta_t$ and $\gamma_t$ do not work well.
- It is difficult to see what figures 1 and 2 refer to.

**Limitations:**

I have no concerns about the limitations and potential negative societal impact.

---

> ### Author Rebuttal · Authors · 2023-08-09
>
> Thank you for your review and your feedback on the notation. We have updated the text, fixing it on the instances that you mentioned. Our response follows.
>
> **On the difficulty of model selection, lower bounds, and minimax optimality.**
> Thank you for this comment. We have added a discussion on lower bounds to the Conclusion section. An overview follows.
>
> Online model selection for bandits is generally perceived to be a hard task with many unresolved problems. There are environments in which model selection may not be possible and the regret is $\Omega(n)$ (cf. Theorem 2 Agarwal et al. 2017). There also exist environments in which the model selection algorithm will perform strictly worse that the algorithm with oracle knowledge, in terms of dependence of the regret on $n$ (cf. Theorem 6.2, Pacchiano et al. 2020). Therefore, the focus of prior so far has been on feasibility, rather than recovering minimax optimal rates. To the best of our knowledge, minimax lower bounds for model selection in linear bandits are *an open problem*. Suppose $B(d,n)$ is the lower bound of the oracle bandit algorithm (for example  $B(d, n) = \Theta(d\sqrt{n})$ when the action domain equals the unit ball). It is *not known* if there exists an algorithm that achieves a $\mathcal{O}(B(d,n)\log M)$ model selection regret. Therefore, we do not know if without further assumptions, the $n^{3/4}$ dependency of our regret bound can be improved or not.
>
> Our work is the *first* to show the feasibility of a $\log M$ upper-bound for model selection in linear bandits, on a general infinite action domain. We are not aware of any prior work which attains a $\mathcal{O}(n^{\alpha}\log M)$ with $\alpha<1$ dependency on general action domains, let alone $\mathcal{O}(\log M\sqrt{n})$.
> To put our contribution in perspective, we mention the results of Foster et al. (2019), which considers model selection for linear bandits over a *finite* action domain of size $K$ and requires knowledge of the horizon $n$. In this setting, they obtain a regret of $$\mathcal{O}\left( \min \Big\[(Mn)^{2/3} (Kd)^{1/3}, K^{1/4}(Mn)^{3/4}+\sqrt{KdMn} \Big\] \right)$$
> which scales polynomially with $M$ and has a potentially suboptimal dependency on $n$.
>
>
> **On computational complexity of ALEXP.**
> The computational complexity of the algorithm scales linearly with $M$, since all agents/models have to be updated. However, updating the models can be done fully in parallel (using tools such as Ray), in which case, increasing the number of models will not affect the runtime of the algorithm. Even without parallelization, the algorithm is light and one complete run of ALEXP with $n=100$, $d=2$, and $M=45$ takes $2.9\pm0.2$ minutes on a single CPU core. Figure 1 in the rebuttal pdf supplement shows how the run-time of our algorithm and other baselines scale with $M$.
>
>
> In general, we suspect that without further assumptions (e.g. relation between $\boldsymbol \phi_j$) linear computational dependency on M may not be avoidable. While algorithms such as CORRAL update only one of the models at every step $t$, their statistical complexity scales polynomially with $M$ (since they require more steps to converge), which results in an overall polynomial computational complexity with $M$.
>
> In the revised paper, we have added Figure 1, and a discussion on computational complexity to Appendix F on “Experiment Details”.
>
>
> **”Algorithm is effective only when M is exponentially large’’**.
> In our experiments, we establish that ALEXP achieves a performance competitive to the oracle, when $M>n$, and $M$ is of the same order of magnitude as $n$.
> Our regret bound demonstrates an upper bound on the worst-case performance of the algorithm. Indeed $n^{3/4}$ is a worse rate than $\sqrt{n}$. However, there are many effective algorithms (e.g. kernelized bandits) who’s (minimax optimal) regret has a worse dependency on the horizon than $\sqrt{n}$. For instance, the commonly used GP-UCB with the $\nu$-Matern kernel satisfies a $\mathcal{O}(n^{\frac{\nu+2d}{2\nu+2d}})$ regret [Whitehouse et al. 2023] which is strictly worse than $\sqrt{n}$.
>
>
> **”Where is the conjectured rate mentioned in Agarwal et al. 2017?”** This can be found in Section 6 of their paper, titled “Conclusion and Open Problems”.
>
>
> Having addressed your concern about rate optimality of the bound, and given the contributions of this paper to the bandit literature, we kindly ask you to reconsider the assessment of our paper. We would be happy to answer any remaining questions or concerns.
>
> ---
> ### References
>
> Agarwal, Alekh, Haipeng Luo, Behnam Neyshabur, and Robert E. Schapire. "Corralling a band of bandit algorithms." In Conference on Learning Theory, pp. 12-38. PMLR, 2017.
>
> Foster, Dylan J., Akshay Krishnamurthy, and Haipeng Luo. "Model selection for contextual bandits." Advances in Neural Information Processing Systems 32 (2019).
>
> Pacchiano, Aldo, My Phan, Yasin Abbasi Yadkori, Anup Rao, Julian Zimmert, Tor Lattimore, and Csaba Szepesvari. "Model selection in contextual stochastic bandit problems." Advances in Neural Information Processing Systems 33 (2020): 10328-10337.
>
> Whitehouse, Justin, Zhiwei Steven Wu, and Aaditya Ramdas. "Improved Self-Normalized Concentration in Hilbert Spaces: Sublinear Regret for GP-UCB." arXiv preprint arXiv:2307.07539 (2023).

---

> > ### Author Response · Authors · 2023-08-16
> > **Rebuttal Follow-up**
> >
> > We hope that our rebuttal has answered your main question about optimality of the bound, in particular your concern about $n^{3/4}$ growth rate being sub-optimal. We emphasize that: the **minimax optimal dependency on horizon is unknown** in this problem setting, and in fact, may not be $\sqrt{n}$. Hao and Lattimore 2020 prove a dimension-independent lower-bound of $\Omega(n^{2/3})$ for *sparse linear bandits* which hints that **going below the $n^{2/3}$ rate might not be possible** for model selection either.
> >
> > We updated the paper and added a short discussion on minimax optimality to sections 5.2. Further, we mentioned the unsolved lower bounds as future work in Section 7.  We hope this update lifts the key concern of your review. It is much appreciated if you could please reconsider your assessment, or respond with questions/suggestions so that we can improve the paper in this regard.
> >
> > ---
> > Hao, Botao, Tor Lattimore, and Mengdi Wang. "High-dimensional sparse linear bandits." Advances in Neural Information Processing Systems 33 (2020): 10753-10763.

---

> > > ### Comment · Reviewer_XgJC · 2023-08-16
> > >
> > > Thank you very much for your kind reply.
> > > I apologize for the delay in responding.
> > > Most of my concerns have been addressed and I am convinced of the importance of this study.
> > > However, I still have concerns about your assertion "Agarwal et al. [2017] raise an open problem on the feasibility of obtaining a log M dependency for the regret.", "We addressed the open problem of Agarwal et al. [2017]".
> > > For details, I would appreciate it if you could check the following:
> > >
> > > In Section 6 of Agarwal et al. [2017], which was mentioned in the authors' responce, I found the following statement in the section you mentioned:
> > > > Another open problem is to improve the dependence on $M$, the number of base algorithms, from polynomial to logarithmic **while keeping the same dependence on other parameters** (or prove its impossibility). **Logarithmic dependence on $M$ can be achieved** by using EXP4 as the master, but as was earlier discussed, this leads to poor dependence on other parameters.
> > >
> > > From this statement, I have the following concerns:
> > > * As long as the dependence on $n$ is worsen, we cannot say that we have addressed the open problem of Agarwal et al. [2017], even if we achieve logarithmic dependence on  $M$.
> > > * Agarwal et al. [2017] believed that one can achieves logarithmi dependence on $M$ by sacrificing the dependence on other parameters.
> > >
> > > If my understanding above was correct, then your assertion that "We addressed the open problem of Agarwal et al. [2017]" would be an exaggeration. I would appreciate a response if there is any misunderstanding regarding this concern.

---

> > > > ### Author Response · Authors · 2023-08-16
> > > >
> > > > Thank you so much for your response. We are glad that your previous questions are answered and are happy to give clarify a few details, which will hopefully raise the current concerns.
> > > >
> > > > **Concern 1.** We do not think that our paper misleads the reader in this regard, as in lines 274-277 we explicitly state that our result matches the conjectured rate of Agarwal et al. only for very large $M$s, and is otherwise suboptimal in $n$. The rest of this paragraph is dedicated to ideas and required assumptions for closing this gap in future work. Nowhere in the paper it is claimed that we resolve the problem of minimax optimal black-box model selection, however we do believe that our work is an important step in this direction (cf. response to concern 2).
> > > >
> > > > In our submission, there is one sentence in the Conclusion section, which mentions *“addressing the open problem of Agarwal et al.”*. We have now changed the verb from “address” to “tackle” to remove misinterpretations and repeated the earlier remark that dependency on $n$ may be suboptimal. We do not intend to exaggerate our results, and would be absolutely happy to further reword this sentence in case you have any suggestions.
> > > >
> > > > **Concern 2.** As you mentioned, Agarwal et al. claim that using EXP4 can yield a $\log M$ rate at the cost of suboptimal dependency on other parameters. **Despite this statement, adapting EXP4 for sample-efficient model selection in bandits is an unresolved challenge.** In fact, we are not aware of any work prior to ours, which presents $\log M$ rates for model selection over general action domains, *even with a suboptimal dependency of $n$*. We give a brief overview of prior attempts to use EXP4 for model selection.
> > > >
> > > > The Exp4 algorithm [Auer et al 2002] was designed for solving $k$-armed bandit problems given the advice of M experts. This algorithm achieves a $\Theta(\sqrt{nk\log M})$ regret w.r.t. the performance of the best-in-hindsight expert, rather than the optimal action $\boldsymbol x^\star$. Moreover, this rate only applies when the experts are static/oblivious and do not adapt to the history $H_t$. Extending this result to 1) learning with *adaptive* agents (which is the case in model selection) and 2) the cumulative regret w.r.t. to $\boldsymbol x^\star$ bears difficulties.
> > > >
> > > > Addressing this problem for *finite action sets* Foster et al. [2019] constructed a variant of EXP4 which roughly put, treated each action as an expert and attained a $\mathrm{poly}(M)$ rate. Improving this, Foster and Rakhlin [2020] later presented SquareCB, which could be used for model selection of linearly parameterizable rewards on finite action sets, and satisfies a $\mathcal{O}(\sqrt{nkd\log M})$ regret [Moradipari et al. 2022]. These attempts use a ridge regression oracle, and naively extending them to infinite action sets will result in a $\sqrt{M}$ dependency. Our results hold for (compact) infinite and non-convex sets. **We are not aware of any other work which applies to such general action sets, and successfully employs EXP4 for model selection.**
> > > >
> > > >
> > > > Please let us know if this answer has not lifted your concerns, and if you have suggestions for modifying the text for further transparency.
> > > > Thank you!
> > > >
> > > > ---
> > > > Auer, Peter, Nicolo Cesa-Bianchi, Yoav Freund, and Robert E. Schapire. "The nonstochastic multiarmed bandit problem." SIAM journal on computing 32, no. 1 (2002): 48-77.
> > > >
> > > > Foster, Dylan J., Akshay Krishnamurthy, and Haipeng Luo. "Model selection for contextual bandits." Advances in Neural Information Processing Systems 32 (2019).
> > > >
> > > > Foster, Dylan, and Alexander Rakhlin. "Beyond ucb: Optimal and efficient contextual bandits with regression oracles." In International Conference on Machine Learning, pp. 3199-3210. PMLR, 2020.
> > > >
> > > > Moradipari, Ahmadreza, Berkay Turan, Yasin Abbasi-Yadkori, Mahnoosh Alizadeh, and Mohammad Ghavamzadeh. "Feature and parameter selection in stochastic linear bandits." In International Conference on Machine Learning, pp. 15927-15958. PMLR, 2022.

---

### Author Rebuttal · Authors · 2023-08-09

We have responded to our reviewers individually. Attached is the pdf supplement, which is referred to in our responses to some of the reviewers.

---

### Decision · Program_Chairs · 2023-09-21

**Decision:**

Accept (poster)

**Comment:**

This paper studies the model selection problem for linear bandits. The main contribution of this paper is an algorithm with a provable regret bound which scales only logarithmically with the number of base models. This is the first such result as all prior works on model selection problems establish regret bounds that are polynomial in the number of models. After the rebuttal and a good discussion all participating reviewers agree that this is a strong paper. One major concern which has been addressed in the rebuttal and discussion is the sub-optimality of the regret bounds in the horizon. It is actually unclear what the min-max regret bound would be both in terms of the number of models and time horizon, under the assumptions made by the authors. This seems like an interesting problem to be addressed in future work. I am happy to recommend this paper for acceptance to the program.